# All-optical geometric image transformations enabled by ultrathin metasurfaces

Xingwang Zhang[1,2], Xiaojie Zhang[1,2], Yao Duan[1], Lidan Zhang[1] & Xingjie Ni ®[1] ✉

Image processing plays a vital role in artificial visual systems, which have diverse applications in areas such as biomedical imaging and machine vision. In particular, optical analog image processing is of great interest because of its parallel processing capability and low power consumption. Here, we present ultra-compact metasurfaces performing all-optical geometric image transformations, which are essential for image processing to correct image distortions, create special image effects, and morph one image into another. We show that our metasurfaces can realize binary image transformations by modifying the spatial relationship between pixels and converting binary images from Cartesian to log-polar coordinates with unparalleled advantages for scale- and rotation-invariant image preprocessing. Furthermore, we extend our approach to grayscale image transformations and convert an image with Gaussian intensity profile into another image with flat-top intensity profile. Our technique will potentially unlock new opportunities for various applications such as target tracking and laser manufacturing.

Geometric transformations are mathematic operations used to modify the geometry of an image by repositioning pixels in a constrained way. In image processing, scale, rotation, and many other geometric transformations are critical steps to correct geometric distortions introduced during the image acquisition processes of image registration, pattern recognition, object tracking, etc[1–3]. In general, geometric image transformation and other image processing techniques are performed digitally, but the speed and power consumption limits of standard image processing chips have become a true bottleneck[4–7]. As such, rapid-growing demands in high-performance machine vision and big data necessitate the advent of novel technologies to carry out fast and energy-efficient image processing. In this context, all-optical analog image processing technologies with massively parallel processing capability have attracted particular attention and hold the promise to solve all those challenges, allowing real-time computation with low power consumption[5,7,8].

All-optical analog image transformations have been fulfilled with conventional optical systems using optical lenses, spatial light modulators[9], and diffractive optical elements[10,11]. These optical image transformation systems offer significant advantages, including real-time operation, parallel processing, and low power consumption.

However, the bulky optical components utilized in these platforms impede further miniaturization and on-chip integration. In addition, conventional diffractive optical elements with microscale addressable pixel sizes suffer from limited spatial resolution and undesirable high-order diffraction loss[10,11]. To fully harness the potential of optical analog image transformations, it is essential to develop a compact optical system that can flexibly process images with high spatial resolution and low loss. Recent advances in dielectric optical metasurfaces have made this possible by enabling remarkably flexible light manipulation on an optically thin layer. These metasurfaces achieve this through the use of engineered subwavelength-sized dielectric nanoantennas, or meta-atoms, which locally impose abrupt changes to optical properties[12–15]. Such structures can manipulate light at the subwavelength scale with minimal loss, thereby offering unparalleled capability for optical analog image processing[5–7,16,17].

Here, we achieved real-time, all-optical geometric image transformations leveraging judiciously designed dielectric metasurfaces (Fig. 1a). Unlike conventional transformation systems that inevitably rely on bulky optical components, our approach utilizes subwavelength-thin flat metasurfaces to realize optical geometric transformations, facilitating potential vertical integration.

[1]Department of Electrical Engineering, The Pennsylvania State University, University Park, PA 16802, USA. [2]These authors contributed equally: Xingwang Zhang, Xiaojie Zhang. ✉e-mail: xingjie@psu.edu

Simultaneously, the subwavelength-scale amorphous silicon meta-atom array that constitutes these flat metasurfaces offers ultra-high spatial resolution and eliminates high-order diffraction loss. To implement geometric image transformations, metasurfaces were designed to perform two-dimensional (2D) space-variant operations by introducing different impulse responses for each pixel of an input image. This allows the input image to be converted into an intentionally distorted image with a modified spatial relationship between pixels.

## Results
### Operation principle
For this purpose, we assume an input image with an amplitude-only transmittance function $f(x, y)$ is projected onto a metasurface with a phase-only transmittance function $e^{i\varphi(x,y)}$ (Fig. 1). There are two sets of phase profiles incorporated in the metasurface $(\varphi(x, y) = \varphi_0(x, y) + \varphi_f(x, y))$: the phase encoded by the desired geometric image transformation $\varphi_0(x, y)$ and the phase of a Fourier

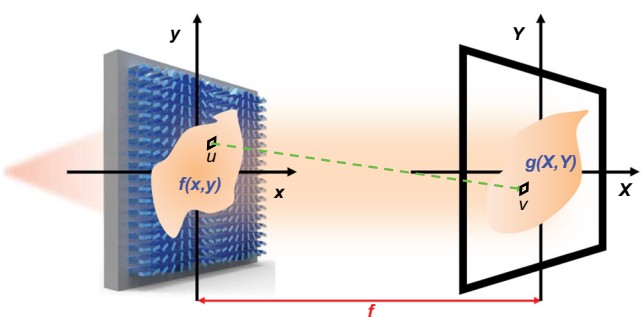

**Fig. 1 | A schematic of geometric image transformation using metasurfaces.** The image of an object is projected on a subwavelength-thin metasurface with nanoscale spatial resolution and is then converted into an intentionally distorted image by repositioning pixels in a constrained manner. The light (the green dotted line) transmitted from each pixel of the image with amplitude-only transmittance $f(x, y)$ in the $(x, y)$ plane is directed by the metasurface with a phase profile $\varphi(x, y)$ to $(X, Y)$ plane, forming an image with a redistributed intensity profile $g(X, Y)$. $u$ and $v$ denote arbitrary pixels in $(x, y)$ and $(X, Y)$ planes, respectively. $f$ is the distance between the input and output planes.

transform lens $\varphi_f(x, y) = -k \cdot \sqrt{x^2 + y^2 + f^2}$, where $k$ is the wave number and $f$ is the focal length. Under the paraxial approximation condition, the corresponding output image $g(X, Y)$ can be described by a modified Fourier transform from the field on the metasurface plane $(x, y)$ to that on the output plane $(X, Y)$ (Supplementary Note 1):

$$g(X, Y) = \iint f(x, y) \cdot e^{i\varphi_0(x, y)} \cdot e^{-ik(x \cdot X + y \cdot Y)/f} dx dy \qquad (1)$$

In the absence of $\varphi_0(x, y)$, the kernel of the integral becomes a Fourier kernel. In this scenario, the metasurface functions as a Fourier transform lens, and $g(X, Y)$ represents the spatial frequency spectrum of the input image $f(x, y)$. In the presence of $\varphi_0(x, y)$, the kernel of the integral turns into a complex term, with the phase being modulated by the additional spatially variant phase distribution $\varphi_0(x, y)$. As a result, the phase term $\varphi_0(x, y)$ incorporated in the Fourier transform metasurface lens can be utilized to geometrically transform the input image $f(x, y)$ into $g(X, Y)$. This can also be understood as follows: for normal light incidence on the metasurface with a phase profile of $\varphi_0(x, y)$, the local light deflection angle in the $(x, z)$ plane can be expressed as $\sin\beta = \frac{1}{k} \cdot \frac{\partial \varphi_0(x,y)}{\partial x}$. The light is subsequently modulated by the Fourier transform phase profile $\varphi_f(x, y)$ and mapped onto the spatial frequency domain $(X, Y)$. Under small angle approximation (Supplementary Note 2)[18], $\sin\beta \approx \tan\beta = \frac{X(x,y)}{f}$. Similarly, in the $(y, z)$ plane, we have $\sin\beta = \frac{1}{k} \cdot \frac{\partial \varphi_0(x,y)}{\partial y} \approx \tan\beta = \frac{Y(x,y)}{f}$. Therefore, we derived the phase gradient of $\varphi_0(x, y)$,

$$\frac{\partial \varphi_0(x,y)}{\partial x} = \frac{k}{f} \cdot X(x, y), \qquad (2)$$

$$\frac{\partial \varphi_0(x,y)}{\partial y} = \frac{k}{f} \cdot Y(x, y). \qquad (3)$$

### Design of metasurfaces
We employed a metasurface consisting of a spatially distributed meta-atom array to apply pixelated phase $\varphi(x, y)$ on the incident light with subwavelength resolution (Fig. 2a, b). Taking advantage of the geometric phase (or Pancharatnam-Berry phase), we utilized amorphous silicon nano-bars with a thickness of 500 nm to convert incident circular polarized light to its orthogonally polarized counterpart with

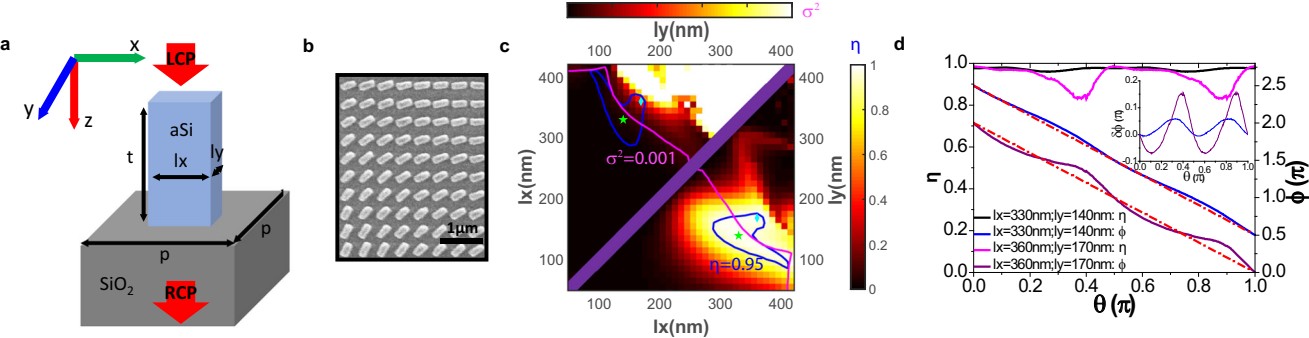

**Fig. 2 | The design principle of meta-atoms. a** A schematic of a unit cell (i.e., meta-atom) of the metasurface consisted of an amorphous silicon (aSi) nanobar on fused silica (SiO₂) substrate. p is the lattice constant. lx, ly and t are the length, width and height of the nanobar, respectively. The left-circular polarized (LCP) light incidence is converted to right-circular polarized (RCP) light accompanied by a certain phase shift, which is determined by the orientation angle of the meta-atom. **b** A scanning electron microscope (SEM) image of a representative metasurface for geometric transformations. The scale bar is 1 μm. **c** The LCP-to-RCP conversion efficiency ($\eta$, lower right region) and phase variance ($\sigma^2$, upper left region) as functions of meta-

atoms dimensions (lx, ly). The blue and pink contour lines indicate $\eta = 0.95$ and $\sigma^2 = 0.001$, respectively. The star and diamond shapes represent two meta-atoms with similar $\eta$ but different $\sigma^2$, located at (330 nm, 140 nm) and (360 nm, 170 nm), respectively. **d** The LCP-to-RCP conversion efficiency ($\eta$) and geometric phase ($\phi$) imparted on the incidence light as functions of meta-atom orientation ($\theta$). Due to the neighboring coupling effects of meta-atoms, $\phi$ deviates from ideal geometric phases with a phase error $\delta\phi$ (the dashed lines). Inset: the phase error $\delta\phi$ as a function of meta-atom orientation ($\theta$) for two meta-atom designs.

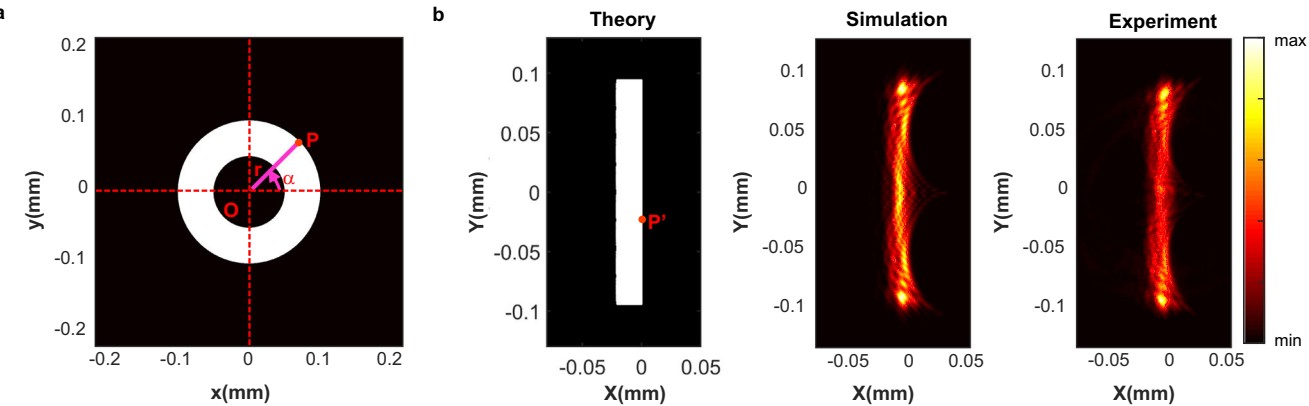

**Fig. 3 | Log-polar coordinate transformation using metasurfaces. a** A ring-shaped object in the Cartesian coordinate with the outer radius of 100 μm and inner radius of 50 μm. The white and black regions indicate transmittance values of 100% and 0%, respectively. The length of OP is $r$, and the angle of OP about $x$-axis is $\alpha$. **b** The analytical (left), simulated (middle), and measurement (right) projections of the object in panel (**a**) from the Cartesian coordinate $(x, y)$ to the log-polar coordinate $(X, Y)$.

specific abrupt phase shifts that are linearly dependent on the nano-bars' orientation angles (Fig. 2a, b). The meta-atom array has a period of 420 nm, allowing only the existence of the zeroth-order diffraction at the operating wavelength of 1064 nm. Consequently, our metasurface-based optical geometric transformer ensures high-order-diffraction-free operations with subwavelength-scale spatial resolution.

To ensure that all the output light carries the required phase, the meta-atoms should be designed to provide nearly unity polarization conversion efficiency. With this goal, we characterized the conversion efficiency ($\eta$) as a function of the nano-bar dimensions ($l_x$, $l_y$). In the pseudo colormap of $\eta$ (Fig. 2c), we can select meta-atom designs within a parameter space where the conversion efficiency exceeds 95%. Ideally, the induced phase shift is twice of the rotation angle of a nano-bar meta-atom. However, due to the inherent field coupling between neighboring meta-atoms, the phase provided by each meta-atom may deviate from the expected value, leading to phase errors defined as the differences between the achieved and required phases (purple curve in Fig. 2d). It is known that in many cases, such as in metalenses, these phase errors can deteriorate the performance of metasurfaces[19,20]. Similarly, we also found that geometric transformations are highly sensitive to phase profile inaccuracies, and the neighboring coupling effect of meta-atoms can significantly degrade the transformation quality (Supplementary Note 3). To address this, we specially engineered our meta-atoms to minimize the neighboring coupling effects. In order to find the optimal designs of the meta-atoms with the minimized neighboring couplings, we calculated the geometric phases ($\phi$) of all meta-atoms as functions of rotation angles ($\theta$), assuming the orientation angles of neighboring meta-atoms are identical. This assumption is reasonable, as the local phase gradient is relatively small in our required phase profiles. Subsequently, we evaluated the phase variance ($\sigma^2$) from the linear regressions of the $\phi - \theta$ plots (Fig. 2c). The phase variance reflects the strength of the meta-atom neighboring coupling; a smaller phase variance indicates a weaker neighboring coupling effect. We identified the meta-atom designs that have the phase variance less than 0.001 (the region below the pink curve in Fig. 2c). Together taking into account the conversion efficiency requirement (*i.e.*, $\eta > 95\%$), we selected the final meta-atom design ($l_x = 330$ nm, $l_y = 140$ nm, marked by a green star) in the intersection region (Fig. 2c). This design has almost unity conversion efficiency ($\eta = 97.7\%$) and negligible neighboring coupling effect ($\sigma^2 = 0.000568$). Compared with other meta-atom designs with similar conversion efficiency (*e.g.*, the one marked by a cyan diamond in Fig. 2c), our chosen meta-atom exhibits significantly better performance on the

optical geometric transformation due to minimized neighboring coupling effects (Supplementary Note 4).

## Geometric transformation for binary images

With the optimized meta-atom design in hand, we employed metasurfaces with minimized neighboring-coupling-induced phase errors to perform geometric image transformations. We initially demonstrated the geometric transformation for binary images (*i.e.*, images whose pixels have two possible intensity values: 0 and 1) and realized Cartesian to log-polar coordinate transformation as an example. In this instance, the metasurface transforms an image with unity transmittance in the Cartesian coordinate to a deformed image in the log-polar coordinate, simultaneously converting the in-plane rotation and scale variations of the input image into translations of the transformed image.

To comprehend the properties of log-polar transformation, we considered an arbitrary point (P) in the Cartesian coordinate $(x, y)$ mapping to point P' in the log-polar coordinate $(X, Y)$ (Fig. 3). Assuming the length of the segment OP is $r$ and the orientation of OP is $\alpha$, the coordinates of P' can be written as $(a \cdot \ln\frac{r}{b}, -a \cdot \alpha)$ with two scale factors $a$ and $b$. For example, mapping a circular ring from the Cartesian into the log-polar coordinates results in a rectangle (Fig. 3b). Furthermore, we can observe that the scale of $r$ and the increment of $\alpha$ are converted to a linear shift along $X$ and $Y$ axes, respectively, in log-polar coordinate. As a result, the original image is transformed into a reformed image that is immune to scale and rotation variations, except for certain translations in the new coordinate.

In order to obtain the encoded phase term $\varphi_0(x, y)$ of metasurfaces for the log-polar coordinate transform, we substituted the coordinate transformation relations $X(x, y) = a \cdot \ln\frac{r}{b}$ and $Y(x, y) = -a \cdot \alpha$ into Eqs. (2, 3). By integrating the spatial phase gradient of $\varphi_0(x, y)$, we can then derive $\varphi_0(x, y) = \frac{k}{f} \cdot [x \cdot X(x, y) + y \cdot Y(x, y) - a \cdot x]$ (Supplementary Note 1).

To experimentally demonstrate log-polar transformation using metasurfaces, we fabricated a metasurface incorporating the geometric transformation encoded phase $\varphi_0(x, y)$ and Fourier transform phase $\varphi_f(x, y)$ (See methods and Supplementary Note 5 for more details on device fabrication). To prepare binary images, we patterned a piece of Aluminum coated glass, which can only allow the light to pass through the transparent region, forming binary images with only two intensity values. To experimentally map an image from Cartesian to log-polar coordinate, we used a 4$f$ system to project the binary image onto our nanofabricated metasurface, and the output transformed image was acquired by a camera through an objective and a

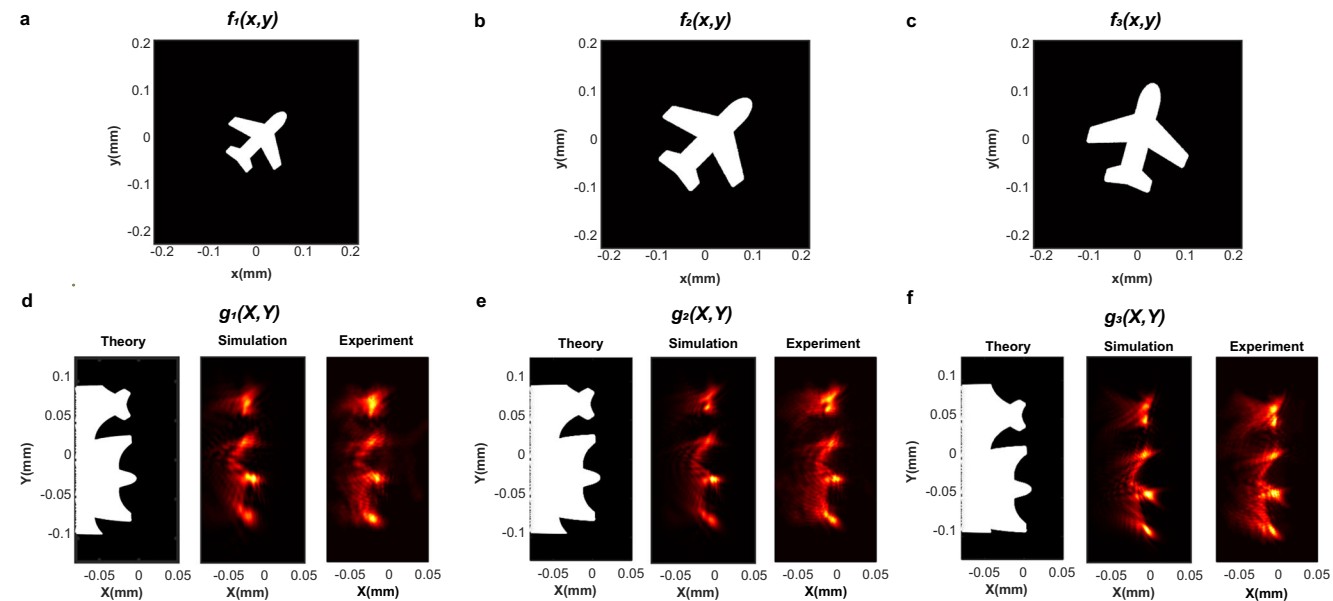

**Fig. 4 | The scale- and rotation-invariance of the log-polar coordinate transformation using metasurfaces.** The log-polar coordinate transformation for input airplane shapes $f(x, y)$ in the Cartesian coordinate with different scale factors ($s$) and rotation angles ($\alpha$). $f_1(x, y)$: $s = 1$ and $\alpha = 0$ (**a**); $f_2(x, y)$: $s = 1.5$ and $\alpha = 0$ (**b**); $f_3(x, y)$: $s = 1.5$ and $\alpha = \pi/6$ (**c**). The white and black regions for the input airplane shapes indicate the transmittance of 100% and 0%, respectively. The theoretical, simulated and experimental transformed images in the log-polar coordinate for the input airplane shapes $f_1(x, y)$ (**d**), $f_2(x, y)$ (**e**) and $f_3(x, y)$ (**f**), respectively.

tube lens (See Methods and Supplementary Note 6 for more details on experimental setup). In our experiment, for the projection of a ring-shaped image on the metasurface, we observed a rectangle image on the camera, which perfectly matched our prediction from simulation (Fig. 3b).

To confirm the scale- and rotation-invariance of our metasurfaces for the log-polar transformation, we utilized three airplane-shaped binary images $f_1(x, y)$, $f_2(x, y)$ and $f_3(x, y)$ with different sizes and orientations as the input images for the metasurface (Fig. 4a–c). As the log-polar transformation is not translation-invariant, and the tolerance for the center position of the airplane shapes is about 10% of the input image's width, we set the center of the shapes as the origin for the Cartesian coordinate to avoid effects from translation (Supplementary Note 11). In our experiment, due to the uneven sampling in $(X, Y)$ domain, the regions further away from the origin in the Cartesian coordinate have higher sampling rate in the log-polar coordinate (Fig. 4d–f, Supplementary Note 7). Therefore, the nose, wings, and the tail of the airplane are brighter in the transformed images. Furthermore, although the input airplane shapes $f_1(x, y)$, $f_2(x, y)$ and $f_3(x, y)$ differed in size and orientation, the transformed images $g_1(X, Y)$, $g_2(X, Y)$ and $g_3(X, Y)$ were nearly identical, with only shifts along $X$ and $Y$ axes (Fig. 4d–f).

To quantitatively characterize the differences between these images, we conducted 2D correlation analysis which is widely used in image processing to evaluate the similarity of two images[21]. We first used the correlation function $R_{f_i f_1} = \mathcal{F}^{-1}\{[\mathcal{F}(f_1)]^* \mathcal{F}(f_i)\}$ to evaluate the similarity between the input images $f_i(x, y)$ and reference image $f_1(x, y)$, where $i = 1$, 2 or 3. $\mathcal{F}$ and $\mathcal{F}^{-1}$ correspond to the Fourier and inverse Fourier transforms. $*$ denotes the complex conjugate. In the case of a perfect match, the autocorrelation function $R_{f_1 f_1}$ between $f_1(x, y)$ and itself displayed a small bright spot in the origin (Fig. 5a). When the airplane was zoomed in by 1.5× (i.e. $s = 1.5$) and then rotated counterclockwise by 30 degrees (i.e. $\alpha = \pi/6$), both the cross-correlation function of test images $f_2(x, y)$ and $f_3(x, y)$ with respect to the reference image $f_1(x, y)$ revealed broad bright regions (Fig. 5b, c), indicating poor similarity. Thus, even though the airplane shapes differed only in size and orientation, their direct correlations failed to recognize that they have the same shape.

In contrast, preprocessing the images with our log-polar transforming metasurfaces effectively addresses this issue. By transforming the images from the Cartesian to log-polar coordinates using our metasurfaces, the scale and rotation of an image are converted into translations in the new coordinate system ($X' = X + a \cdot \ln s$, $Y' = Y - a \cdot \alpha$, Supplementary Note 8). According to the shift theorem of the correlation function $R_{g_1 g_i}(X + X_0, Y + Y_0) = \mathcal{F}^{-1}\{[\mathcal{F}(g_1(X, Y))]^* \mathcal{F}(g_i(X + X_0, Y + Y_0))\}$, the translation of $g_i$ can shift the correlation $R_{g_1 g_i}$ with the same amount without deforming the appearance of $R_{g_1 g_i}$. Therefore, by performing 2D correlation analysis, we can not only accurately recognize the test images, but also quantitatively determine their scale factors and rotation angles.

To showcase the superior properties of our metasurfaces for the scale- and rotation-invariant image processing, we also used the correlation function to characterize the similarity between the three experimentally transformed airplane images in Fig. 4d, e, f. The auto-correlation map $R_{g_1 g_1}$ of $g_1(X, Y)$ was obtained for comparison, where a bright spot can be observed at the origin, indicating a perfect match (Fig. 5d). There are two more weak spots on each side along the $Y$ axis due to the similar features of the transformed image along the polar axis (Fig. 4d). Next, we obtained the cross-correlation maps $R_{g_1 g_2}$ and $R_{g_1 g_3}$ for the transformed airplane images $g_2(X, Y)$ and $g_3(X, Y)$ with respect to $g_1(X, Y)$, which have a scale factor and/or a rotation angle, respectively. In contrast to the poor similarity indicated by the direct cross-correlations without preprocessing (Fig. 5b, c), we observed an excellent match between the transformed images, indicated by the small bright spots (Fig. 5e, f). For the airplane image with a scale factor of $s = 1.5$ (Fig. 4b), the corresponding cross-correlation map of its transformed image has the brightest spot at (12.32 μm, 0 μm), which can be translated to a scale factor of 1.5 (Fig. 5e, $X_0 = a \cdot \ln s$, and $a = 30$ μm). Similarly, for the transformed airplane image with both a rotation angle of $\alpha = 30$ degrees and a scale factor of $s = 1.5$ (Fig. 4c), the brightest correlation spot located at (12.32 μm, −16.27 μm) (Fig. 5f). The offsets in both axes indicate the test airplane image was rotated by 31 degrees and zoomed by 1.5×, which agrees well with the actual values ($Y_0 = -a \cdot \alpha$, and $a = 30$ μm).

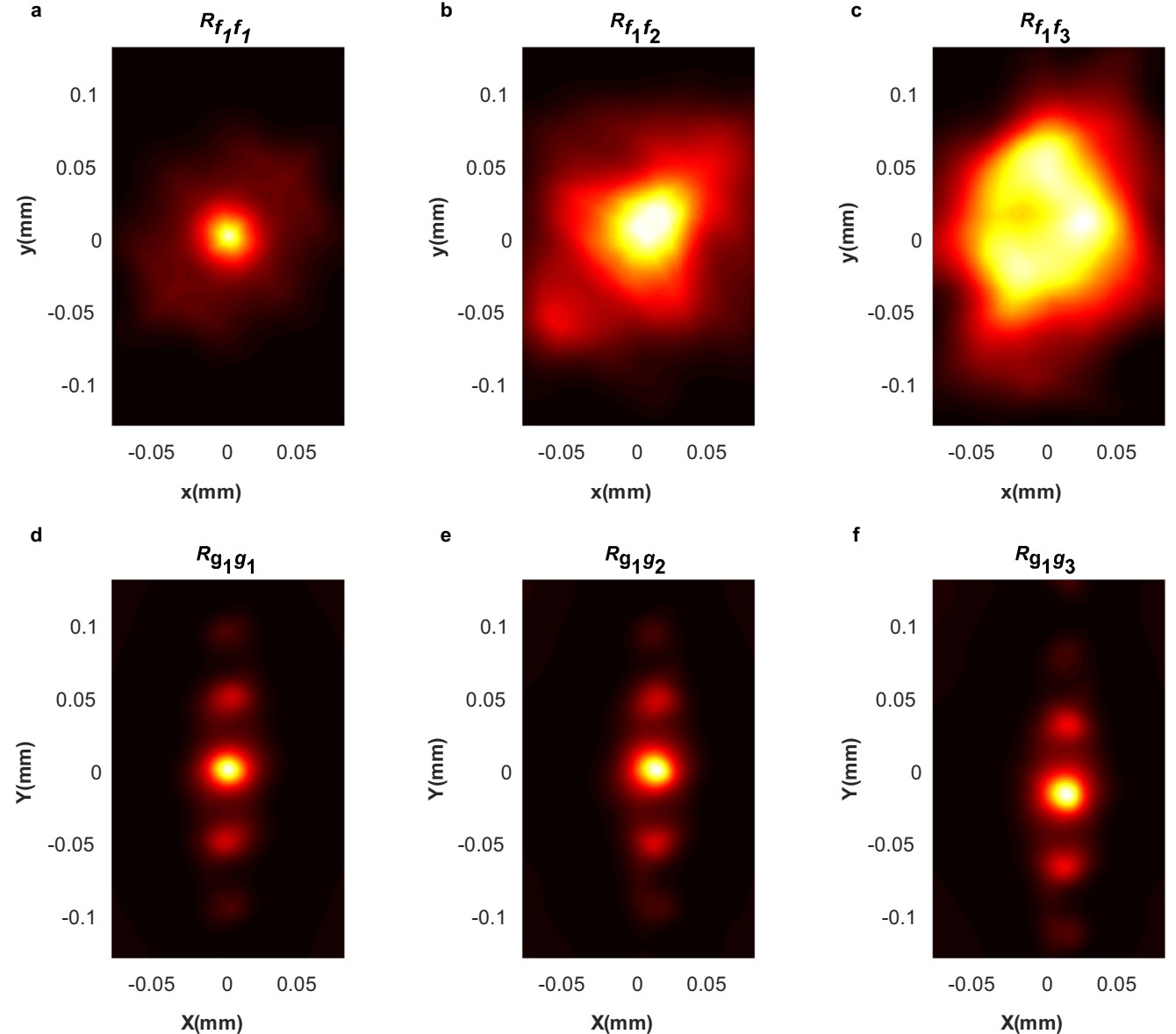

**Fig. 5 | The correlation comparison between images in Cartesian and log-polar coordinates.** The correlations for the input airplane shapes $f_i(x, y)$ in the Cartesian coordinate (**a**–**c**). The correlation function is $R_{f_1 f_i} = \mathcal{F}^{-1}\{[\mathcal{F}(f_1)]^* \mathcal{F}(f_i)\}$, where $i = 1, 2$ or 3. The correlations for the transformed images $g_i(X, Y)$ in the log-polar coordinate (**d**–**f**). The correlation function is $R_{g_1 g_i} = \mathcal{F}^{-1}\{[\mathcal{F}(g_1)]^* \mathcal{F}(g_i)\}$, where $i = 1, 2$ or 3.

It is important to note that in practical situations, the acquired image of an object typically varies in size and/or orientation depending on its relative position with the imaging system. Therefore, it is desirable and advantageous to transform images in advance to make them resistant to scale and rotation in applications such as pattern recognition, target tracking, and image registration. Our geometric image-transforming metasurfaces have the potential to be applied in image preprocessing for these fields. In addition, we can also transform the log-polar image which was previously transformed from an image in the Cartesian coordinate, back to the original image by a log-polar to Cartesian coordinate transforming metasurface (Supplementary Note 9). Furthermore, although our metasurfaces can only work for the log-polar coordinate transformation of binary images, it is also possible to transform color images by using metasurfaces operating at red, green, and blue wavelengths (Supplementary Note 10)[22].

## Geometric transformation for grayscale images

So far, we have demonstrated the exceptional advantages of metasurfaces for scale- and rotation-invariant transformation of binary images by considering only their local geometric characteristics. We proceeded to extend our metasurfaces for the transformation of grayscale images with both intensity distribution and local geometric information. To this end, we designed a metasurface to perform transformation from a grayscale image with circular Gaussian profile to a square flat-top shape (i.e., Gaussian-to-flat-top transformation). To obtain the transformation encoded phase $\varphi_0(x, y)$ for the metasurface, we considered a transformation from a Gaussian profile with $f(x, y) = \exp\left[-\frac{2(x^2 + y^2)}{r_0^2}\right]$ to a square flat-top one with $g(X, Y) = \frac{1}{4w_0^2} \cdot \mathrm{rect}\left(\frac{X}{2w_0}\right) \cdot \mathrm{rect}\left(\frac{Y}{2w_0}\right)$. In accordance with the optical geometric transformation principle, we can divide both the input Gaussian and the output square flat-top profiles into $N$ parts with an

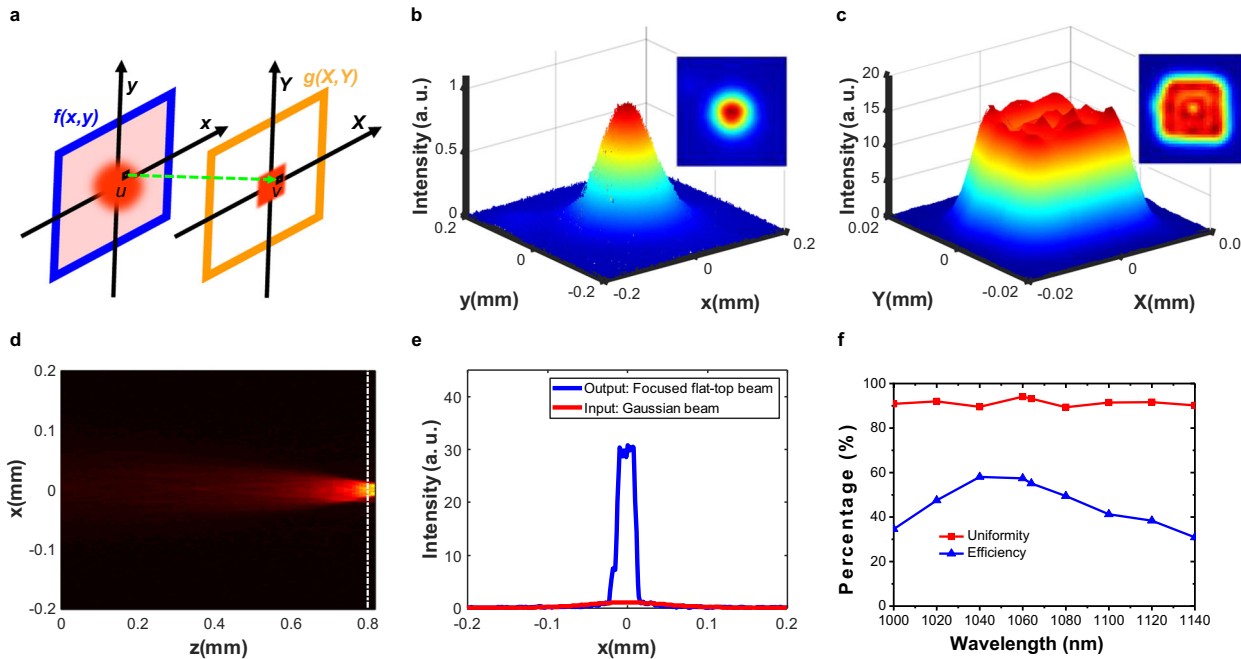

**Fig. 6 | Gaussian-to-flat-top geometric transformation using metasurfaces. a** A schematic for field mapping from the Gaussian beam at $(x, y)$ plane to the flat-top beam at $(X, Y)$ plane. According to the energy conservation low, an arbitrary pixel $u$ of the Gaussian beam is mapped to pixel $v$ of the flat-top beam with the same energy. **b** The measured intensity profile for a Gaussian beam with rotational symmetry. The waist diameter of the beam is 200 μm. **c** The measured intensity profile for the transformed square-shaped flat-top beam. The size of the square beam is 20 μm. **d** The evolution of the intensity distribution along z-axis for the Gaussian-to-flat-top beam transformation from metasurface plane ($z = 0$) to the focal plane ($z = 800$ μm) (the dashed white line). **e** The comparison of input Gaussian and output flat-top beam profiles. **f** The flat-top beam uniformity and transformation efficiency as functions of input wavelength.

equal amount of energy. An arbitrary small piece $u$ in the $(x, y)$ plane is then redirected by the metasurface to $v$ in $(X, Y)$ plane (Fig. 6a). Based on the energy conservation law, the energy in $u$ and $v$ should be equal, resulting in $\int_u f(x,y)du = \int_v g(X,Y)dv$.[23] Therefore, for the mapping from $(x, y)$ to $(X, Y)$, we have $\iint_{x,y=0}^{x,y} f(x,y)dxdy = \iint_{X,Y=0}^{X,Y} g(X,Y)dXdY$, which yields the coordinate transformation relationships $X = w_0 \cdot \text{erf}\left(\sqrt{2}\frac{|x|}{r_0}\right)$ and $Y = w_0 \cdot \text{erf}\left(\sqrt{2}\frac{|y|}{r_0}\right)$, where $\text{erf}(\xi)$ is error function and is defined as $\text{erf}(\xi) = \frac{2}{\sqrt{\pi}}\int_0^\xi \exp(-\xi^2)d\xi$. Combing the above coordinate relationships with Eqs. (2) and (3), we can finally obtain the required transformation phase profile by integrating the spatial phase gradient of $\varphi_0(x, y)$, leading to $\varphi_0(x,y) = 2\sqrt{2\pi}\frac{r_0 w_0}{f\lambda} \cdot \left\{ \left[ \frac{\sqrt{\pi}}{2} \cdot \xi_x \cdot \text{erf}(\xi_x) + \frac{1}{2} \cdot \exp(-\xi_x^2) - \frac{1}{2} \right] + \left[ \frac{\sqrt{\pi}}{2} \cdot \xi_y \cdot \text{erf}(\xi_y) + \frac{1}{2} \cdot \exp(-\xi_y^2) - \frac{1}{2} \right] \right\}$, where $\xi_x = \sqrt{2}\frac{|x|}{r_0}$ and $\xi_y = \sqrt{2}\frac{|y|}{r_0}$ (Supplementary Note 1).

To experimentally realize the Gaussian-to-flat-top transformation with a metasurface, we fabricated a metasurface with the required phase profile encoded (See Methods and Supplementary Note 5 for more details on device fabrication). To characterize the performance of the fabricated metasurface, we prepared the Gaussian grayscale image with a Gaussian laser beam with a waist diameter of 200 μm (See Methods and Supplementary Note 6 for more details on experimental setup), and the metasurface was placed on the beam waist to ensure normal incidence (Fig. 6b). After passing through our metasurface, the beam was transformed into a square-shaped flat-top cross-section with a beam width of $2w_0 = 20 \mu m$ at the focal plane of the metasurface (Fig. 6c). To gain insight into the field evolution during the transformation process, we measured the light intensity distribution for cross-sections of the beam at different distances away from the metasurface plane (Fig. 6d, e), the Gaussian beam was gradually focused into a square spot with enhanced light intensity from the metasurface plane to

the focal plane. In contrast to existing metalenses, which focus the incident beam into a small spot with a circular-shaped Gaussian intensity distribution, our geometric transformation metasurface provides a square-shaped spot with both the intensity flatness and edge sharpness (Fig. 6e).

The importance of generating an optical beam with a square shape, uniform intensity, and sharp edges in laser processing technology cannot be overstated, as these characteristics are crucial for obtaining high-quality micro-structures with enhanced efficiency, reduced surface roughness, and sharper sidewalls[24,25]. Compared to other methods that use lenses or diffractive optical elements to generate a flat-top beam, our metasurface-based flat-top beam transformer offers advantages with its ultra-compact platform, subwavelength-scale thickness, superior spatial resolution, and inherent diffraction-loss-free property. This makes it a highly enticing solution for applications such as laser drilling, scribing, and welding.

We also characterized the operation bandwidth of our flat-top transformation metasurfaces. To achieve this, we captured the output flat-top beam profiles at different input laser wavelengths using a camera and subsequently calculated the corresponding uniformities and efficiencies. As demonstrated in Fig. 6f, the uniformities of the transformed flat-top beams in the spectral range between $\lambda = 1000$ nm and $\lambda = 1140$ nm remain consistently around 90%. To calculate the transformation efficiency, we divided the integrated intensity inside the square-shaped region by the total integrated intensity of the input Gaussian beam. As illustrated in Fig. 6f, the transformation efficiencies between $\lambda = 1000$ nm and $\lambda = 1020$ nm are all above 35%, with the peak efficiency reaching up to 58% between $\lambda = 1040$ nm and $\lambda = 1060$ nm.

## Discussion
In conclusion, we have successfully demonstrated geometric image transformations using subwavelength-thin all-dielectric metasurfaces with minimized neighboring-coupling-induced phase errors. Our geometric transformation metasurfaces are capable of handling both

binary and grayscale images, exhibiting remarkable performance in terms of low optical loss and high spatial resolution. By achieving Cartesian to log-polar coordinate transformation, we unlock unprecedented potential for scale- and rotation-invariant image processing, crucial for applications in pattern recognition, target tracking, and image registration. Moreover, we demonstrated grayscale image transformations using metasurfaces by converting a Gaussian beam into a flat-top one, potentially paving the way for applications in high-precision laser manufacturing and opening new possibilities in laser drilling, scribing, and welding. Furthermore, the planar nature of our metasurfaces allows for the potential vertical integration of multiple metasurfaces, resulting in even more sophisticated optical functionalities. Our technology is poised to have a significant impact on various industries, providing versatile and efficient solutions to a wide range of applications in optical data processing.

## Methods

### Device fabrication

To fabricate metasurfaces for geometric image transformation, we first deposited a layer of amorphous silicon film with a thickness of 500 nm on a fused silica substrate by Plasma Enhanced Chemical Vapor Deposition (PECVD). Then we spin-coated e-beam resist on top of the silicon film followed by e-beam lithography. After development, we deposited a thin layer of Aluminum on the sample as the hard mask by e-beam evaporation for the dry etching followed by a lift-off process. Then we transferred the pattern onto the silicon layer by using Inductively Coupled Plasma - Reactive Ion Etching (ICP-RIE). Finally, we removed the residual Aluminum layer by wet-etching method (Supplementary Note 5). To fabricate the test images for log-polar coordinate transformation, we first deposited a layer of Aluminum film with a thickness of 100 nm on a glass slide by e-beam evaporation. The transmittance for 100 nm-thick Aluminum film is about $10^{-6}$ around $\lambda = 1064$ nm, which is small enough to block the incident laser beam. Then we spin-coated a layer of photo-resist on top of Aluminum film. After that, we used a laser writer to expose the patterns of the test images on the photo-resist. After development, we transferred the pattern onto the Aluminum film by using ICP-RIE. As such, the transparent region of the patterned Aluminum film formed the test images for log-polar coordinate transformation (Supplementary Note 5).

### Experimental setup

We utilized a collimated laser beam ($\lambda = 1064$ nm) as the light source and used a linear polarizer and a quarter wave plate to prepare the circular polarization state to meet the requirement of the metasurface. We then used a 4× objective (NA = 0.2) and a tube lens to project the transformed images on a camera. For the binary image transformation, the collimated laser illuminated the patterned Aluminum coated glass slide to prepared the test images which were scaled down by 4× by a 4$f$ system and projected on the metasurface plane. For the grayscale image transformation, the laser was coupled by a single-mode fiber to ensure single-mode Gaussian beam output. The Gaussian beam was then expanded by a beam expander and projected on the metasurface plane (Supplementary Note 6).

## Data availability

All data is available in the main text and the supplementary information. The data that support the findings of this study are available from the corresponding author upon reasonable request.

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

## Acknowledgements

The work was partially supported by the Moore Inventor Fellow award from the Gordon and Betty Moore Foundation, the National Aeronautics and Space Administration Early Career Faculty Award (80NSSC17K0528), the Office of Naval Research Basic Research Challenge (N00014-18-1-2371), the National Eye Institute of the National Institutes of Health (1R21EY031853-01), and the NSF CAREER Award (ECCS-2047446).

## Author contributions

X.W.Z., X.J.Z. and X.N. conceived the project. X.W.Z. and X.J.Z. conducted metasurfaces design, simulations, and experiments. X.W.Z., X.J.Z. and X.N. analyzed the data and wrote the paper. Y.D. and L.Z. fabricated the devices. X.W.Z. provided technical support for the device fabrication. X.N. supervised the study.

## Competing interests

The authors declare no competing interests.
