## [Peer Review File · Nature Communications]

All-optical geometric image transformations enabled by ultrathin metasurfacesREVIEWER COMMENTS

Reviewer #1 (Remarks to the Author):

Metasurface based all-optical image processing (e.g., edge imaging) has been a hot topic recently. Zhang and colleagues demonstrated 2D geometric image transformations based on the conversion from Cartesian to log-polar coordinates. They also extended this approach to 3D geometric image transformations. Although the results are interesting, I have hesitation to recommend its publication before the following issues and concerns can be well addressed.

1. There are several works on real-time 2D image transformation. For instance, 1. Zokai, Siavash, and George Wolberg. "Image registration using log-polar mappings for recovery of large-scale similarity and projective transformations." *IEEE Transactions on Image Processing* 14.10 (2005): 1422-1434. 2. Zhang, Yi, et al. "A novel quantum representation for log-polar images." *Quantum information processing* 12 (2013): 3103-3126. 3. Wolberg, G., & Zokai, S. (2000, September). Robust image registration using log-polar transform. In *Proceedings 2000 International Conference on Image Processing (Cat. No. 00CH37101)* (Vol. 1, pp. 493-496). IEEE. Please compare their work with these works.

2. Fig.1a is very misleading since the grayscale 3D image in it is not demonstrated in the manuscript. I am a little bit confused with the key information in Fig.1b. What information does the authors want to let readers know by showing the dotted green lines meet in the middle? I would suggest authors to revise this figure by providing more information.

3. In Fig. 4, design (theoretical) and simulation results should be added for the sake of comparison with the experimental result, like the one shown in Fig. 3. Moreover, Fig. 4 should be labelled properly for better readability.

4. On Page 4, lines 175-177 and Supplementary Section 5, the authors mentioned that four bright regions in the transformed image are due to the uneven sampling at corners of the imaging object. However, the object shown in Figure 4 has more than four corners (e.g., the alone tail has four corners). A detailed explanation is needed.

5. How to understand cross-correlation as shown in Fig 4(g-l). How were they calculated? I think the intensity profiles in Fig 4(g-l) don't provide enough information.

6. How did they extend this to 3D image transformation? No formulae in the manuscript give the information about the 3rd coordinate. More detailed derivation process should be provided in the Supplementary.

7. It looks like that the demonstration of Gaussian-to-flat-top geometric transformation is a kind of beam shaping, which can be easily realized with currently available method (https://www.rp-photonics.com/flat_top_beams.html). It is not very convincing to claim that this is a proper 3D image transformation. It would be great if the authors can demonstrate a proper 3D object (e.g., 3D plane in the current Fig.1a).

8. A real time image may contain grayscale or colour information with a complex intensity profile. It would be great if authors can also discuss the design limitations in the Discussion section.

9. Overall writing should be improved. There are several grammar issues (e.g, Page 8, Line 143-145) and different fonts (Page 8, Line 193-196).

Reviewer #2 (Remarks to the Author):

The manuscript by Zhang et al. proposes a metasurface-based all-optical geometric image transformation that plays a vital role in various fields of image processing, such as pattern recognition, target tracking, and image registration. Specifically, with the aid of the designed metasurface, the transformed input image into the spatial frequency domain is invariant to scaling and rotation, which is highly useful for the image classification. Compared to conventional spatial light modulator or diffractive optical element-based optical analog image transformations, the proposed metasurface suppresses undesirable high-order diffraction orders, and increases spatial resolution within a miniaturized form factor. In addition, the authors have specially engineered meta-atoms to ensure that output light carries the designed phase by optimizing the geometric parameters of meta-atoms which minimize the neighboring coupling effects between adjacent meta-atoms. The authors provide a clear and thorough explanation of the simulations and the experimental results. The paper is timely and should be of great interest to a broad range of communities including optics, computing, and information processing. Therefore, I believe this manuscript deserves publication in Nature Communications once some questions are properly addressed by the author.

1. The importance of log-polar transformation in image classification is well explained. It is understood that the metasurface conducting log-polar transformation can work with various input images. I recommend the authors to comment on any conditions regarding the input images, such as their physical size, and how these conditions may affect the design of the metasurface.
2. For image classification, it seems that the metasurface operates in the visible wavelength regime with at least three wavelengths, RGB, being preferred. In addition to the discussion on scale- and rotation-invariant properties of the proposed metasurfaces, I recommend the authors to add a discussion on the wavelength-dependent variance and the resulting offset on the transformed plane.
3. In Figure S3, the effect of neighboring coupling between adjacent meta-atoms is discussed by comparing analytical and FDTD results. However, this analysis was conducted on nanodisks. I recommend the authors to compare star and diamond rectangular meta-atoms, as denoted in Fig. 2C of the main manuscript, using FDTD simulations to emphasize the effectiveness of the specially engineered meta-atoms.

Reviewer #3 (Remarks to the Author):

The authors report optical image transformation using all-dielectric metasurface. They intend to produce large changes in optical images; one of the examples is Cartesian to log-polar coordinate transformation. The design principle is similar to metalens and holography using metasurfaces, adjusting phase of light finely. As far as I examined this manuscript (MS), the most practical application is to produce flat top beam (Fig. 5).

To consider this MS further, the following points should be clarified.

- 1) Definitions are unclear.
Lines 79,80: k , f are not defined explicitly.
Line 93: θ is defined as deflection angle. However, θ is assigned to rotation angle in line 125.
- 2) Small angle approximation
Line 95: Small angle approximation is used. But there is no justification.
Actual range of θ should be noted.

3) Fig. 3, 4

Actual optical and SEM images of the metasurfaces are highly preferred to be shown.
In Fig. 3, how was the break of the ring implemented in the metasurface structure?

4) Fig. 4

Is it possible to reproduce the original plane image based on Fig. 4d,g,j
(or e,h,k or f,i,l)?

If yes, how?

If no, why?

5) Log-polar to Cartesian

It is informative for readers to see optical transformation from log-polar to Cartesian coordinate. Could the authors show the example(s)?

At least, the procedure(s) to realize them is expected to be described.

6) missing information

Height of Si nanostructure, defined as t , is not shown.

Ref. 4 does not have volume and page.

Point-by-Point Response Letter

We thank the reviewers for their valuable feedback. Each comment is in red, which is addressed directly and as clearly as possible in blue. All the changes we made are highlighted in the main text and are included here in green. In addition, we have revised the Supplementary Information to make it more readable, and the references to the Supplementary Information in the main text have been modified accordingly.

Responses to Reviewer 1:

Metasurface based all-optical image processing (e.g., edge imaging) has been a hot topic recently. Zhang and colleagues demonstrated 2D geometric image transformations based on the conversion from Cartesian to log-polar coordinates. They also extended this approach to 3D geometric image transformations. Although the results are interesting, I have hesitation to recommend its publication before the following issues and concerns can be well addressed.

We greatly appreciate the reviewer's generally positive feedback and insightful comments on our work. Due to these constructive comments, our revised manuscript has been significantly enhanced. We have incorporated additional theoretical and simulation results in the revised Fig. 4 (in response to Comment #1.3), broadened our theoretical analysis on geometric image transformations using metasurfaces in the newly added Supplementary Notes 1-2 (in response to Comment #1.6), included more experimental data on the vertical integration of image transformation metasurfaces (in response to Comment #1.7), and appended design and simulation data for metasurfaces for multi-color geometric image transformations in the newly added

Supplementary Note 10 (in response to Comment #1.8).

In the following sections, we will address the reviewer's concerns in a point-by-point manner to provide clarity on our interpretation of our results. But first, we would like to highlight the significance of our work as outlined below:

1. In processes like image registration, pattern recognition, and object tracking within image processing, geometric image transformations play a critical role in correcting geometric distortions introduced during image acquisition processes. In our work, we achieved all-optical image geometric transformation that offers **significant advantages such as fast speed, low power consumption, and a massively parallel processing capability** compared to digital image processing. Furthermore, utilizing optical metasurfaces to perform image transformation not only enhances our approach but also **substantially reduces device thickness to a subwavelength scale, paving the way for potential vertical integration**.
2. Our approach also be applied for grayscale image transformations by adjusting the intensity profile of grayscale images. As a demonstration, we converted an input Gaussian beam into a focusing beam with a flat-top profile using a metasurface, achieving both high efficiency and excellent uniformity. **Our metasurfaces-based Gaussian-to-flat-top geometric transformation stands out from conventional techniques due to its advantages such as subwavelength-scale device thickness, enhanced spatial resolution, inherent diffraction-loss-free properties, and the potential for vertical integration**.
3. In the revised manuscript, **we have extended our method to achieve multi-wavelength image transformation**, and demonstrated that color images can be transformed using metasurfaces operating at red, green, and blue wavelengths.
4. **Our approach can also be expanded to other types of geometric**

transformations. As verification, we conducted additional studies on the log-polar to Cartesian coordinate transformation using metasurfaces and included the corresponding findings in the revised manuscript.

As a summary, our manuscript presents a generalized method for all-optical geometric image transformation using the subwavelength-thin metasurfaces, establishing a desired coordinate transformation relationship between the input and output images. We have demonstrated that our metasurfaces can **work for both binary and grayscale images** in the main text and **extend to color images** in the revised Supplementary Information.

1.1. There are several works on real-time 2D image transformation. For instance, 1. Zokai, Siavash, and George Wolberg. "Image registration using log-polar mappings for recovery of large-scale similarity and projective transformations." *IEEE Transactions on Image Processing* 14.10 (2005): 1422-1434. 2. Zhang, Yi, et al. "A novel quantum representation for log-polar images." *Quantum information processing* 12 (2013): 3103-3126. 3. Wolberg, G., & Zokai, S. (2000, September). Robust image registration using log-polar transform. In *Proceedings 2000 International Conference on Image Processing (Cat. No. 00CH37101) (Vol. 1, pp. 493-496)*. IEEE. Please compare their work with these works.

We appreciate the reviewer's insightful comment and for highlighting these three studies. Each of these papers showcases critical applications of 2D image transformation technology, particularly in image registrations. To elucidate the uniqueness and significance of our research, we provide a comparison with these three works below.

Paper #1 introduces **Cartesian-to-log-polar coordinates transformation algorithm in the spatial domain**, serving as a preprocessing module to recover large-scale changes and arbitrary rotations in digital image registration.

Paper #2 presents a **fast rotation-invariant quantum image registration algorithm for log-polar images** and demonstrates that the algorithm can achieve a quartic speedup. Paper #3 describes a **hierarchical image registration algorithm** designed for affine motion recovery, utilizing the log-polar registration module to accommodate arbitrary rotation angles and a wide range of scale changes.

As we can see, all three papers focused on the “**algorithm development**” for log-polar geometric transformation using **digital image processing technology**. In contrast, our work realizes *all-optical image geometric transformation using analog image processing technology*, boasting significant advantages such as fast speed, low power consumption, and a massively parallel processing capability. Furthermore, utilizing optical metasurfaces to perform image transformation not only enhances our approach but also substantially reduces device thickness to a subwavelength scale, paving the way for potential vertical integration.

Given that the references recommended by the reviewer provide strong evidence for the applications of geometric image transformation, we have cited them as Ref. 1, 2 and 3 in the first paragraph on Page 3 in the revised main text, which now reads:

“

... .. In image processing, scale, rotation, and many other geometric transformations are critical steps to correct geometric distortions introduced during the image acquisition processes of image registration, pattern recognition, object tracking, etc^{1,2,3}.... ..

1. Zokai S, Wolberg G. Image registration using log-polar mappings for recovery of large-scale similarity and projective transformations. IEEE Trans Image Process 14, 1422-1434 (2005).

2. Zhang Y, Lu K, Gao Y, Xu K. A novel quantum representation for log-polar images. Quantum Inf Process 12, 3103-3126 (2013).

3. Wolberg G, Zokai S. Robust image registration using log-polar

transform. Proceedings 2000 International Conference on Image Processing 1, 493-496 (2000).

”

1.2. Fig.1a is very misleading since the grayscale 3D image in it is not demonstrated in the manuscript. I am a little bit confused with the key information in Fig.1b. What information does the authors want to let readers know by showing the dotted green lines meet in the middle? I would suggest authors to revise this figure by providing more information.

We thank the reviewer for pointing it out. In our work, we first demonstrated the Cartesian to log-polar coordinate transformation with binary images, wherein pixels comprise two intensity levels, and realized as an example. In this case, we only considered the local geometric characteristics of the images. Then, **we expanded the application of our metasurfaces for the transformation of *grayscale images* with both intensity distribution and local geometric information.** To demonstrate, we utilized a metasurface to perform transformation from a grayscale image with circular Gaussian profile to a uniform square flat-top shape. In Fig. 1b of the previous version's main text, dotted green lines were used to describe the optical field mapping from x-y plane to the X-Y plane.

To prevent potential misunderstandings, we have revised Fig. 1 and its caption in the revised manuscript to more clearly illustrate the principle of grayscale geometric image transformation showcased in our work. For the reviewer's convenience, we have also included the relevant revised sections below for the reviewer's convenience.

Fig. 1 now reads:

“

Fig. 1. A schematic of geometric image transformation using metasurfaces. The image of an object is projected on a subwavelength-thin metasurface with nanoscale spatial resolution and is then converted into an intentionally distorted image by repositioning pixels in a constrained manner. The light (the green dotted line) transmitted from each pixel of the image with amplitude-only transmittance $f(x, y)$ in the (x, y) plane is directed by the metasurface with a phase profile $\varphi(x, y)$ to (X, Y) plane, forming an image with a redistributed intensity profile $g(X, Y)$.

”

1.3. In Fig. 4, design (theoretical) and simulation results should be added for the sake of comparison with the experimental result, like the one shown in Fig. 3. Moreover, Fig. 4 should be labelled properly for better readability.

We appreciate the reviewer’s suggestions. We have now added both the theoretical and simulation results in Fig. 4. We also have added the label for each panel. For better readability, we have restructured Fig. 4 and moved the correlation analysis part to Fig. 5.

Fig. 4 now reads:

“

Fig. 4. The scale- and rotation-invariance of the log-polar coordinate transformation using metasurfaces. The log-polar coordinate transformation for input airplane shapes $f(x, y)$ in the Cartesian coordinate with different scale factors (s) and rotation angles (α). $f_1(x, y)$: $s=1$ and $\alpha=0$ (a); $f_2(x, y)$: $s=1.5$ and $\alpha=0$ (b); $f_3(x, y)$: $s=1.5$ and $\alpha=\pi/6$ (c). The white and black regions for the input airplane shapes indicate the transmittance of 100% and 0%, respectively. The mathematical, simulated and experimental transformed images in the log-polar coordinate for the input airplane shapes $f_1(x, y)$ (d), $f_2(x, y)$ (e) and $f_3(x, y)$ (f), respectively.

”

Figure 5 now reads:

“

Fig. 5. The correlation comparison between images in Cartesian and log-polar coordinates. The correlations for the input airplane shapes $f_i(x, y)$ in the Cartesian coordinate (a-c). The correlation function is $R_{f_i f_i} = \mathcal{F}^{-1}\{[\mathcal{F}(f_i)]^* \mathcal{F}(f_i)\}$, where $i=1, 2$ or 3 . The correlations for the transformed images $g_i(X, Y)$ in the log-polar coordinate (d-f). The correlation function is $R_{g_i g_i} = \mathcal{F}^{-1}\{[\mathcal{F}(g_i)]^* \mathcal{F}(g_i)\}$, where $i=1, 2$ or 3 .

”

Furthermore, corresponding changes have been made to the main text. The updated last paragraph on page 9 of the main text reads as follows:

“

To confirm the scale- and rotation-invariance of our metasurfaces for the log-polar transformation, we utilized three airplane-shaped binary images $f_1(x, y)$, $f_2(x, y)$ and $f_3(x, y)$ with different sizes and orientations as the input images for the metasurface (**Figs. 4a-4c**). In our experiment, due to the uneven sampling in (X, Y) domain, the regions further away from the origin in the Cartesian coordinate have higher sampling rate in the log-polar coordinate (Figs. 4d-4f, Supplementary Note 7). Therefore, the nose,

wings, and the tail of the airplane are brighter in the transformed images.

Furthermore, although the input airplane shapes $f_1(x,y)$, $f_2(x,y)$ and $f_3(x,y)$ differed in size and orientation, the transformed images $g_1(X,Y)$, $g_2(X,Y)$ and $g_3(X,Y)$ were nearly identical, with only shifts along X and Y axes (Figs. 4d-4f).

”

The second paragraph on page 10 in the main text now reads as follows:

“

To quantitatively characterize the differences between these images, we conducted 2D correlation analysis which is widely used in image processing to evaluate the similarity of two images.²¹ We first used the correlation function $R_{f_i, f_i} = \mathcal{F}^{-1}\{\mathcal{F}(f_i) \mathcal{F}^*(f_i)\}$ to evaluate the similarity between the test images $f_i(x,y)$ and reference image $f_1(x,y)$, where $i=1, 2$ or 3 . \mathcal{F} and \mathcal{F}^{-1} correspond to the Fourier and inverse Fourier transforms. $*$ denotes the complex conjugate. In the case of a perfect match, the autocorrelation function R_{f_1, f_1} between $f_1(x,y)$ and itself displayed a small bright spot in the origin (Fig. 5a). When the airplane was zoomed in by $1.5\times$ (i.e. $s=1.5$) and then rotated counterclockwise by 30 degrees (i.e. $\alpha=\pi/6$), both the cross-correlation function of test images $f_2(x,y)$ and $f_3(x,y)$ with respect to the reference image $f_1(x,y)$ revealed broad bright regions (Figs. 5b and 5c), indicating poor similarity. Thus, even though the airplane shapes differed only in size and orientation, their direct correlations failed to recognize that they have the same shape.

”

The last paragraph on page 10 in the main text now reads as follows:

“

In contrast, preprocessing the images with our log-polar transforming metasurfaces effectively addresses this issue. By transforming the images from

the Cartesian to log-polar coordinates using our metasurfaces, the scale and rotation of **an image** are converted into translations in the new coordinate system **$(X' = X + a \cdot \ln s, Y' = Y - a \cdot \alpha, \text{Supplementary Note 8})$** . **According to the shift theorem of the correlation function $R_{g_1 g_i}(X + X_0, Y + Y_0) = \mathcal{F}^{-1}\{[\mathcal{F}(g_1(X, Y))]^* \mathcal{F}(g_i(X + X_0, Y + Y_0))\}$, the translation of g_i can shift the correlation $R_{g_1 g_i}$ with the same amount without deforming the appearance of $R_{g_1 g_i}$. Therefore, by performing 2D correlation analysis, we can not only accurately recognize the test images, but also quantitatively determine their scale factors and rotation angles.**

”

1.4. On Page 4, lines 175-177 and Supplementary Section 5, the authors mentioned that four bright regions in the transformed image are due to the uneven sampling at corners of the imaging object. However, the object shown in Figure 4 has more than four corners (e.g., the alone tail has four corners). A detailed explanation is needed.

We thank the reviewer for pointing out the error. Based on the log-polar coordinate transformation relationship, an image at the Cartesian coordinate (x, y) is mapped to the log-polar coordinate (X, Y) using $X(x, y) = a \cdot \ln \frac{r}{b}$ and $Y(x, y) = -a \cdot [\text{atan2}(y, x)]$. In our experiment, the image was uniformly illuminated and the meta-atoms were arranged in a square lattice, both of which determined the uniform spatial sampling rate in the (x, y) domain. However, in the log-polar coordinate (X, Y) , the sampling rate along X axis is proportional to the inverse of r ($\Delta X = a \frac{\Delta r}{r}$). **Therefore, regions further away from the origin in the Cartesian system exhibit a higher sampling rate (and thus, higher grayscale intensity) in the log-polar system.** Given that the airplane’s nose, wings, and tail are distanced from the origin in the Cartesian system, their

corresponding transformed regions in the log-polar coordinate are notably brighter.

We have revised the relevant section. The last paragraph on page 9 in the revised main text now reads:

“
... ..In our experiment, due to the uneven sampling in (X,Y) domain, the regions further away from the origin in the Cartesian coordinate have higher sampling rate in the log-polar coordinate (Figs. 4d-4f, Supplementary Note 7). Therefore, the nose, wings, and the tail of the airplane are brighter in the transformed images.
”

1.5. How to understand cross-correlation as shown in Fig 4(g-l). How were they calculated? I think the intensity profiles in Fig 4(g-l) don't provide enough information.

We thank the reviewer for the comment. The 2D correlation function is a standard tool in image processing to evaluate the similarity between two images. If the 2D correlation exhibits a bright tiny spot, it means a perfect similarity between two images (Figs. 4g, 4j, 4k and 4l in the previous version of main text). In contrast, the larger spot in the 2D correlation map indicates the worse similarity (Figs. 4h and 4i in the previous version of main text).

Considering the airplane shapes $f_1(x,y)$, $f_2(x,y)$, and $f_3(x,y)$ in Figs. 4a-4c, each differs in size and orientation. To assess the similarity between $f_i(x,y)$ and $f_1(x,y)$, where $i=1, 2$ or 3 , the correlation function can be expressed as $R_{f_1f_i} = \mathcal{F}^{-1}\{[\mathcal{F}(f_1)]^* \mathcal{F}(f_i)\}$. Here \mathcal{F} and \mathcal{F}^{-1} represent the Fourier and inverse Fourier transforms, and $*$ denotes the complex conjugate. **The autocorrelation $R_{f_1f_1}$ reveals a bright tiny spot, indicating perfect similarity between $f_1(x,y)$ and itself.** Yet, although the airplane shapes

$f_1(x, y)$, $f_2(x, y)$, and $f_3(x, y)$ differed only in size and orientation, both $R_{f_1 f_2}$ and $R_{f_1 f_3}$ reveal large bright spots. **Hence, the 2D correlation function is ineffective for image recognition when the images are subject to changes in scale and rotation.**

In contrast, preprocessing these images with our log-polar transforming metasurface effectively addressed this issue. By transforming the images $f_1(x, y)$, $f_2(x, y)$, and $f_3(x, y)$ from Cartesian coordinate to $g_1(X, Y)$, $g_2(X, Y)$, and $g_3(X, Y)$ in log-polar coordinate using our metasurface, changes in the object's scale and rotation translate to mere displacements. **Consequently, both the cross-correlation $R_{g_1 g_2}$ and $R_{g_1 g_3}$ display bright small spots, indicating that the 2D correlation function remains effective with log-polar transformation.**

Analytically, if we consider an image $f(r, \theta)$, where $r = \sqrt{x^2 + y^2}$ and $\theta = \text{atan2}(y, x)$ in Cartesian coordinate system, in the log-polar coordinate system, the image is transformed into $g(X, Y)$, where $X = a \cdot \ln \frac{r}{b}$ and $Y = -a \cdot \theta$. When the image is scaled by s and rotated by α (i.e. $f'(r', \theta') = f(sr, \theta + \alpha)$), we can obtain $X' = X + a \cdot \ln s$ and $Y' = Y - a \cdot \alpha$ in the log-polar coordinate system. Therefore, the scale- and rotation- changes in the Cartesian coordinate become linear translations in the log-polar coordinate.

For the correlation function $R_{g_1 g_i}(X, Y) = \mathcal{F}^{-1}\{\left[\mathcal{F}(g_1(X, Y))\right]^* \mathcal{F}(g_i(X, Y))\}$, we have $R_{g_1 g_i}(X + X_0, Y + Y_0) = \mathcal{F}^{-1}\{\left[\mathcal{F}(g_1(X, Y))\right]^* \mathcal{F}(g_i(X + X_0, Y + Y_0))\}$. **Thus, the linear translations $g_i(X + X_0, Y + Y_0)$ in the log-polar coordinate shift the center of the correlation $R_{g_1 g_i}(X + X_0, Y + Y_0)$ with the same amount.** In summary, **for the images with scale and rotation differences, the correlation calculations for their log-polar transformations can not only display small bright spots, but also transform scale and rotation changes into linear translations.**

To address this important point, we have added the detailed correlation analysis on pages 10-11 in the main text. We have included the relevant sections below for the reviewer's convenience.

The second and third paragraphs on page 10 in the main text now reads:

“

To quantitatively characterize the differences between these images, we conducted 2D correlation analysis which is widely used in image processing to evaluate the similarity of two images.²¹ We first used the correlation function $R_{f_1 f_i} = \mathcal{F}^{-1}\{[\mathcal{F}(f_1)]^* \mathcal{F}(f_i)\}$ to evaluate the similarity between the test images $f_i(x, y)$ and reference image $f_1(x, y)$, where $i=1, 2$ or 3 . \mathcal{F} and \mathcal{F}^{-1} correspond to the Fourier and inverse Fourier transforms. $*$ denotes the complex conjugate. In the case of a perfect match, the autocorrelation function $R_{f_1 f_1}$ between $f_1(x, y)$ and itself displayed a small bright spot in the origin (Fig. 5a). When the airplane was zoomed in by $1.5\times$ (i.e. $s=1.5$) and then rotated counterclockwise by 30 degrees (i.e. $\alpha=\pi/6$), both the cross-correlation function of input images $f_2(x, y)$ and $f_3(x, y)$ with respect to the reference image $f_1(x, y)$ revealed broad bright regions (Figs. 5b and 5c), indicating poor similarity. Thus, even though the airplane shapes differed only in size and orientation, their direct correlations failed to recognize that they have the same shape.

In contrast, preprocessing the images with our log-polar transforming metasurfaces effectively addresses this issue. By transforming the images from the Cartesian to log-polar coordinates using our metasurfaces, the scale and rotation of an image are converted into translations in the new coordinate system ($X' = X + a \cdot \ln s$, $Y' = Y - a \cdot \alpha$, Supplementary Note 8). According to the shift theorem of the correlation function $R_{g_1 g_i}(X + X_0, Y + Y_0) = \mathcal{F}^{-1}\{[\mathcal{F}(g_1(X, Y))]^* \mathcal{F}(g_i(X + X_0, Y + Y_0))\}$, the translation of g_i can shift the correlation $R_{g_1 g_i}$ with the same amount without deforming the

appearance of $R_{g+\theta_i}$. Therefore, by performing 2D correlation analysis, we can not only accurately recognize the test images, but also quantitatively determine their scale factors and rotation angles.

”

1.6. How did they extend this to 3D image transformation? No formulae in the manuscript give the information about the 3rd coordinate. More detailed derivation process should be provided in the Supplementary.

We apologize for any confusion our terminology may have caused. By “2D geometric image transformation,” we refer to the transformation applied to a binary image. In contrast, the term “3D geometric image transformation” describes the geometric transformation for grayscale images, which can be represented in the format (x, y, I) , with I indicating grayscale intensity. To prevent future misunderstandings, we have changed the terms in the revised manuscript: “2D image transformation” has been replaced with “binary image transformation,” and “3D image transformation” is now “grayscale image transformation”.

Moreover, to provide a clearer understanding of the derivation process for the geometric image transformation, we have restructured this section and included additional details in the revised Supplementary Note 1. For the reviewer’s convenience, the newly added Supplementary Note 1 is provided below.

“

1. The theory for geometric image transformation using metasurfaces

Figure S1. A schematic for field mapping from (x, y) plane to the focal plane (X, Y) of a metasurface. Under normal incidence, the light transmitted from each pixel of an image with amplitude-only transmittance $f(x, y)$ in the (x, y) plane is directed by the metasurface with a phase profile $\varphi(x, y)$ to (X, Y) plane, forming an image with a redistributed intensity profile $g(X, Y)$.

We assume an input image with an amplitude-only transmittance function $f(x, y)$ is projected onto a metasurface with a phase-only transmittance function $e^{i\varphi(x, y)}$, and then transformed to an image $g(X, Y)$. As illustrated in Fig. S1, the phase profile $\varphi(x, y)$ for the metasurface can be expressed as:

$$\varphi(x, y) = \varphi_0(x, y) + \varphi_f(x, y), \quad (\text{S1})$$

where $\varphi_0(x, y)$ is the phase encoded by the desired geometric image transformation, and $\varphi_f(x, y)$ is the phase of a Fourier transform lens, and we have

$$\varphi_f(x, y) = -k \cdot \sqrt{x^2 + y^2 + f^2}, \quad (\text{S2})$$

where, f is the focal length and k is the wave number. Under the paraxial approximation condition, the corresponding output image $g(X, Y)$ can be

described by a modified Fourier transform from the field on the metasurface plane (x, y) to that on the output plane (X, Y)

$$g(X, Y) = \iint f(x, y) \cdot e^{i\varphi_0(x, y)} \cdot e^{-ik(x \cdot X + y \cdot Y)/f} dx dy \quad (\text{S3})$$

In the absence of $\varphi_0(x, y)$, the kernel of the integral becomes a Fourier kernel. In this scenario, the metasurface functions as a Fourier transform lens, and $g(X, Y)$ represents the spatial frequency spectrum of the input image $f(x, y)$. In the presence of $\varphi_0(x, y)$, the kernel of the integral turns into a complex term, with the phase being modulated by the additional spatially variant phase distribution $\varphi_0(x, y)$. As a result, the phase term $\varphi_0(x, y)$ incorporated in the Fourier transform metasurface lens can be utilized to geometrically transform the input image $f(x, y)$ into $g(X, Y)$. This can also be understood as follows: for normal light incidence on the metasurface with a phase profile of $\varphi_0(x, y)$, the local light deflection angle in the (x, z) plane can be expressed as

$$\sin \beta = \frac{1}{k} \cdot \frac{\partial \varphi_0(x, y)}{\partial x}. \quad (\text{S4})$$

The light is subsequently modulated by the Fourier transform phase profile $\varphi_f(x, y)$ and mapped onto the spatial frequency domain (X, Y) . Under small angle approximation,¹ we have

$$\sin \beta \approx \tan \beta = \frac{X(x, y)}{f}. \quad (\text{S5})$$

Similarly, in the (y, z) plane, we have

$$\sin \beta = \frac{1}{k} \cdot \frac{\partial \varphi_0(x, y)}{\partial y} \approx \tan \beta = \frac{Y(x, y)}{f}. \quad (\text{S6})$$

Therefore, we derived the phase gradient of $\varphi_0(x, y)$,

$$\frac{\partial \varphi_0(x,y)}{\partial x} = \frac{k}{f} \cdot X(x,y), \quad (\text{S7})$$

$$\frac{\partial \varphi_0(x,y)}{\partial y} = \frac{k}{f} \cdot Y(x,y). \quad (\text{S8})$$

We first consider only the geometric characteristics of the image $f(x,y)$ and neglected the grayscale information. In this case, the image $f(x,y)$ can be expressed as:

$$f(x,y) = \begin{cases} 1, & x,y \in A \\ 0, & x,y \notin A \end{cases} \quad (\text{S9})$$

where, A is the outline of the image. For the log-polar coordinate transform of an image, we have the coordinate relations as below:

$$X(x,y) = a \cdot \ln \frac{r}{b}, \quad (\text{S10})$$

$$Y(x,y) = -a \cdot \alpha, \quad (\text{S11})$$

$$r = \sqrt{x^2 + y^2}, \quad (\text{S12})$$

$$\alpha = \text{atan2}(y,x), \quad (\text{S13})$$

where a and b are scale factors. In order to obtain the encoded phase term $\varphi_0(x,y)$ of metasurfaces for the log-polar coordinate transform, we substituted the coordinate transformation relations Eqs. S10-S13 into Eqs. S7-S8. By integrating the spatial phase gradient of $\varphi_0(x,y)$, we can then derive

$$\varphi_0(x,y) = \frac{k}{f} \cdot [x \cdot X(x,y) + y \cdot Y(x,y) - a \cdot x]. \quad (\text{S14})$$

Therefore, combing Eq. S1, Eq. S2 and Eq. S14, we can obtain the phase profile $\varphi(x,y)$ of metasurfaces for log-polar coordinate transformation:

$$\varphi(x,y) = \frac{k}{f} \cdot [x \cdot X(x,y) + y \cdot Y(x,y) - a \cdot x] - k \cdot \sqrt{x^2 + y^2 + f^2}. \quad (\text{S15})$$

For the log-polar coordinate transformation, we considered only the geometric characteristics of the image $f(x,y)$ and neglected the grayscale information. Therefore, the metasurfaces for log-polar coordinate transformation only works for binary images whose pixels have only two possible intensity values (Eq. S9).

To transform grayscale images with both intensity distribution and local geometric information, we need to incorporate both grayscale and geometric information of the image $f(x, y)$. To this end, we can divide both the input image $f(x, y)$ and the output image $g(X, Y)$ into N parts with an equal amount of energy. An arbitrary small piece u in the (x, y) plane is then redirected by the metasurface to v in (X, Y) plane (Fig. S1). Based on the energy conservation law, the energy in u and v should be equal, resulting in $\int_u f(x, y) du = \int_v g(X, Y) dv$.² Therefore, for the mapping from (x, y) to (X, Y) , we have the expression

$$\iint_{x,y=0}^{x,y} f(x, y) dx dy = \iint_{X,Y=0}^{X,Y} g(X, Y) dX dY. \quad (\text{S16})$$

As an example, we considered a transformation from a Gaussian grayscale profile to a square flat-top profile:

$$f(x, y) = \exp\left[-\frac{2(x^2+y^2)}{r_0^2}\right], \quad (\text{S17})$$

$$g(X, Y) = \frac{1}{4w_0^2} \cdot \text{rect}\left(\frac{X}{2w_0}\right) \cdot \text{rect}\left(\frac{Y}{2w_0}\right), \quad (\text{S18})$$

where r_0 is the radius of the Gaussian profile and w_0 is the half of the width of the flat-top profile. By substituting Eqs. S17 and S18 into Eq. S16, we can derive the coordinate transformation relationships:

$$X = w_0 \cdot \text{erf}\left(\sqrt{2}\frac{|x|}{r_0}\right), \quad (\text{S19})$$

$$Y = w_0 \cdot \text{erf}\left(\sqrt{2}\frac{|y|}{r_0}\right), \quad (\text{S20})$$

where $\text{erf}(\xi)$ is error function and is defined as $\text{erf}(\xi) = \frac{2}{\sqrt{\pi}} \int_0^\xi \exp(-\xi^2) d\xi$.

Combing the Eqs. S19-S20 with Eqs. S7 and S8, we can finally obtain the required transformation phase profile by integrating the spatial phase gradient of $\varphi_0(x, y)$, leading to

$$\varphi_0(x, y) = 2\sqrt{2\pi} \frac{r_0 w_0}{f\lambda} \cdot \left\{ \left[\frac{\sqrt{\pi}}{2} \cdot \xi_x \cdot \text{erf}(\xi_x) + \frac{1}{2} \cdot \exp(-\xi_x^2) - \frac{1}{2} \right] + \left[\frac{\sqrt{\pi}}{2} \cdot \xi_y \cdot \text{erf}(\xi_y) + \frac{1}{2} \cdot \exp(-\xi_y^2) - \frac{1}{2} \right] \right\}, \quad (\text{S21})$$

where $\xi_x = \sqrt{2} \frac{|x|}{r_0}$ and $\xi_y = \sqrt{2} \frac{|y|}{r_0}$. Therefore, combining Eq S1, Eq. S2 and Eq. S21, we can obtain the phase profile $\varphi(x, y)$ of metasurfaces for Gaussian-to-flat-top transformation:

$$\varphi(x, y) = 2\sqrt{2\pi} \frac{r_0 w_0}{f\lambda} \cdot \left\{ \left[\frac{\sqrt{\pi}}{2} \cdot \xi_x \cdot \text{erf}(\xi_x) + \frac{1}{2} \cdot \exp(-\xi_x^2) - \frac{1}{2} \right] + \left[\frac{\sqrt{\pi}}{2} \cdot \xi_y \cdot \text{erf}(\xi_y) + \frac{1}{2} \cdot \exp(-\xi_y^2) - \frac{1}{2} \right] \right\} - k \cdot \sqrt{x^2 + y^2 + f^2}. \quad (\text{S22})$$

In this work, the phase profiles used for Cartesian to log-polar and Gaussian-to-flat-top transformations are plotted in Fig. S2.

Figure S2. The phase profiles for Cartesian to log-polar (a) and Gaussian-to-flat-top (b) transformations.

”

We have also included the citation of the newly added Supplementary Note 1 in the main text. The first paragraph on page 5 in the main text now reads:

“

... .. Under the paraxial approximation condition, the corresponding output

image $g(X, Y)$ can be described by a modified Fourier transform from the field on the metasurface plane (x, y) to that on the output plane (X, Y) **(Supplementary Note 1):**

“

The second paragraph on page 9 in the main text now reads:

“

... .. By integrating the spatial phase gradient of $\varphi_0(x, y)$, we can then derive $\varphi_0(x, y) = \frac{k}{f} \cdot [x \cdot X(x, y) + y \cdot Y(x, y) - a \cdot x]$ **(Supplementary Note 1).**

“

The first paragraph on page 13 in the main text now reads:

“

... .. Combing the above coordinate relationships with Eqs. (2) and (3), we can finally obtain the required transformation phase profile by integrating the spatial phase gradient of $\varphi_0(x, y)$, leading to $\varphi_0(x, y) = 2\sqrt{2\pi} \frac{r_0 w_0}{f \lambda} \cdot \left\{ \left[\frac{\sqrt{\pi}}{2} \cdot \xi_x \cdot \operatorname{erf}(\xi_x) + \frac{1}{2} \cdot \exp(-\xi_x^2) - \frac{1}{2} \right] + \left[\frac{\sqrt{\pi}}{2} \cdot \xi_y \cdot \operatorname{erf}(\xi_y) + \frac{1}{2} \cdot \exp(-\xi_y^2) - \frac{1}{2} \right] \right\}$, where $\xi_x = \sqrt{2} \frac{|x|}{r_0}$ and $\xi_y = \sqrt{2} \frac{|y|}{r_0}$ **(Supplementary Note 1).**

“

1.7. It looks like that the demonstration of Gaussian-to-flat-top geometric transformation is a kind of beam shaping, which can be easily realized with currently available method (https://www.rp-photonics.com/flat_top_beams.html). It is not very convincing to claim that this is a proper 3D image transformation. It would be great if the authors can demonstrate a proper 3D object (e.g., 3D plane in the current Fig.1a).

We thank the reviewer for the comment. We again apologize for the confusion caused by our terminology. We have changed “3D image

transformation” to “grayscale image transformation” (see our response to Comment #1.6).

As shown in the link mentioned by the reviewer and Refs.23 and 25 in our revised main text, Gaussian-to-flat-top geometric transformation has been previously realized by using a lens system and diffractive optical elements. However, **the bulky nature of lens systems impedes further miniaturization and on-chip integration**. Although diffractive optical elements can reduce the device thickness to tens of microns, **they suffer from limited spatial resolution and undesirable high-order diffraction loss**. In addition, **the fabricating these diffractive optical elements requires complicated multi-step lithography to construct pixels with different thicknesses**.

In contrast, the metasurfaces in our work can achieve Gaussian-to-flat-top geometric transformation in a much more compact way due to their ultrathin formfactor. The subwavelength-scale meta-atoms that constitute these metasurfaces offer ultra-high spatial resolution and eliminates high-order diffraction loss. Furthermore, the planar nature of our metasurfaces allows for the potential vertical integration of multiple metasurfaces, resulting in even more sophisticated optical functionalities. **Therefore, our metasurface-based Gaussian-to-flat-top geometric transformation significantly outperforms conventional techniques due to its subwavelength-scale device thickness, high spatial resolution, inherent diffraction-loss-free property, and vertical integration for extended functionality.**

We have added the comparison with other methods using lenses and diffractive optical elements in the main text to emphasize the unique advantages of our metasurfaces for Gaussian-to-flat-top geometric transformation.

The second paragraph on page 14 in the main text now reads:

“

... ..Compared to other methods that use lenses or diffractive optical elements to generate a flat-top beam, our metasurface-based flat-top beam transformer

offers advantages with **its ultra-compact platform, subwavelength-scale thickness, superior spatial resolution, and inherent diffraction-loss-free property**. This makes it a highly enticing solution for applications such as laser drilling, scribing, and welding.

”

In addition, we can exploit the planar nature of our metasurfaces to realize vertical integration (*i.e.*, integration along z axis), aiming at more sophisticated optical functionalities on Gaussian-to-flat-top geometric transformation.

To verify the vertical integration capability of our metasurfaces for new functionalities, we conducted additional experiments. As can be seen in Fig. R1 below, we integrated another layer of metasurface on the opposite side of the fused silica. The front layer of the metasurface is utilized to transform the geometric shape and intensity distribution, while the back layer of the metasurface is used to compensate the phase distortion, such that the final output beam is expected to have a square-shaped flat-top intensity profile with uniform phase distribution for collimated propagation. It should be noted that collimated flat-top beams are of importance to microscopy illumination and MOPA lasers, where long working ranges are required.

Bearing this in mind, we designed a doublet metasurface to transform a Gaussian beam with a radius of $r_0=200\mu\text{m}$ to a square flat-top beam with a width of $2w_0=400\mu\text{m}$ as a proof-of-concept demonstration. In order to present a clear comparison, we first studied the beam evolution with the front metasurface only. As can be seen in Fig. R2a and 2b, the input Gaussian beam gradually evolved into a flat-top beam at the back surface of the fused silica chip (*i.e.* $z=1$ mm). Without phase correction, the flat-top beam diverged quickly after passing through the back side of the fused silica chip with a deterioration of the beam uniformity (Fig. R2a). In our experiment, as shown in Figs. R2d and R2e, we also observed that the beam exhibited a quick divergence behavior.

To achieve collimated flat-top beam propagation, we added another layer of metasurface on the back side of the chip for phase correction. We calculated the flat-top beam evolution after passing through the second layer metasurface. In contrast to the quick divergence shown in Fig. R2a, the beam for the doublet metasurface maintained uniform for 2 mm propagation (Fig. R2c). We also fabricated the second layer metasurface in experiment for phase correction. As can be seen in Fig. R2f, we observed the collimated flat-top beam was collimated as long as 2 mm propagation.

Figure R1. Integrated metasurface doublet for collimated Gaussian-to-flat-top laser beam transformation. (a) The schematic for the integrated metasurface doublet. Two metasurfaces are fabricated on both sides of a fused silica with a thickness of 1mm. (b) The phase profile for Metasurface #1. (c) The phase profile for Metasurface #2.

Figure R2. Demonstration of collimated Gaussian-to-flat-top beam transformation using an integrated metasurface doublet. The simulated (a) and measured (d) beam evolution from the first layer metasurface ($z=0$ mm) to $z=3$ mm without the second layer of metasurfaces. The dashed white lines indicate the back surface of the fused silica chip. The simulated (b) and measured (c) intensity profiles for the input Gaussian beam (top) and the output flat-top beam (bottom). The simulated (e) and measured (f) beam evolution from the second layer of metasurface ($z=1$ mm) to $z=3$ mm for the metasurface doublet.

1.8. A real time image may contain grayscale or colour information with a complex intensity profile. It would be great if authors can also discuss the design limitations in the Discussion section.

We thank the reviewer for this excellent suggestion. In our main text, we have demonstrated that our metasurfaces can realize transformations of binary images by modifying the spatial relationship among pixels and converting binary images from Cartesian to log-polar coordinates, offering unparalleled advantages for scale- and rotation-invariant image preprocessing (Figs. 3-4).

Furthermore, **we have also extended our approach to transformations of grayscale images by redistributing the intensity profile of these images.** As a demonstration, we transformed an input Gaussian beam into a focusing beam with a flat-top profile with high efficiency and excellent uniformity by a metasurface (Fig. 6).

For images containing color information (i.e., wavelength information), we have conducted additional studies. **First**, we analyzed the phase profiles for log-polar coordinate transformation and found that the image would be distorted due to chromatic phase aberration when the operational wavelength changed. **Then**, we simulated the transformation of a ring-shaped image at different wavelengths. The results revealed that the transformed images became bent rectangles because of the chromatic aberration.

To overcome the chromatic aberration, we designed a new set meta-atoms that works at three distinct wavelengths (red, green, and blue). We incorporated the phase profile $\varphi(x, y)$ for each of these wavelengths separately to avoid the chromatic aberration. **Finally**, we performed the log-polar coordinate transformation for the same ring-shaped image. Unlike the significant chromatic aberration observed with previous metasurface design, the transformed images using the new RGB metasurface remained consistent across red, green, and blue colors. Therefore, we have demonstrated that our metasurfaces can operate for RGB color images.

We have now added the corresponding contents in the revised Supplementary Note 10, and included below.

“

10. Metasurfaces for multi-color geometric image transformations

From Eq. S15, we know the phase profile $\varphi(x, y)$ of metasurfaces for log-polar coordinate transformation is wavelength dependent. Therefore, we can re-write it as

$$\varphi(x, y, \lambda) = \frac{2\pi}{\lambda} \cdot \left\{ \frac{1}{f} \cdot [x \cdot X(x, y) + y \cdot Y(x, y) - a \cdot x] - \sqrt{x^2 + y^2 + f^2} \right\}. \quad (\text{S31})$$

For simplicity, we can obtain

$$\varphi(x, y, \lambda) = \frac{2\pi}{\lambda} \cdot \Phi(x, y), \quad (\text{S32})$$

$$\Phi(x, y) = \frac{1}{f} \cdot [x \cdot X(x, y) + y \cdot Y(x, y) - a \cdot x] - \sqrt{x^2 + y^2 + f^2}, \quad (\text{S33})$$

where $\Phi(x, y)$ is wavelength independent. In our work, the metasurfaces were design at $\lambda = 1064 \text{ nm}$. Therefore, if we change the wavelength of the incident laser, the transformed image will be distorted as a result of the chromatic phase aberration (Eq. S32).

On the other hand, the transmittance (i.e. LCP-to-RCP conversion efficiency) for our meta-atom was also wavelength dependent. As shown in Fig. S17, our meta-atom has a peak transmittance of about 98% at $\lambda=1064 \text{ nm}$. When the wavelength deviates from $\lambda=1064 \text{ nm}$, the transmittance drops down to about 60% at $\lambda=980 \text{ nm}$ and $\lambda=1200 \text{ nm}$. Therefore, if we change the wavelength of the incident laser, the transformed image will become dim due to smaller transmittance efficiency.

Figure S17. LCP-to-RCP conversion efficiency for our meta-atoms at different wavelengths.

To further study the wavelength dependent characteristics for our log-polar coordinate transformation, we simulated the transformation for the ring-shaped image in Fig. 3a in the main text at different wavelengths, and the results are

plotted in Fig. S18. When the wavelength deviates from $\lambda=1064$ nm, the transformed images turn dim due to smaller transmittance. In addition, when the wavelength is beyond ± 20 nm away from $\lambda=1064$ nm, the transformed images become bended rectangles due to the chromatic aberration (Eq. S32)

Figure S18. The log-polar coordinate transformed images using a metasurface at different wavelengths.

To overcome the chromatic aberration, we need to incorporate the wavelength dependence characteristics for the phase profile in Eq. S32. To this end, we can design the meta-atoms working at three different wavelengths (e.g. Red, Green and Blue), and design the phase profile $\varphi(x, y)$ for the three different wavelengths separately to avoid the chromatic aberration. As such we can obtain

$$\varphi(x, y, \lambda_R) = \frac{2\pi}{\lambda_R} \cdot \Phi(x, y), \quad (\text{S34})$$

$$\varphi(x, y, \lambda_G) = \frac{2\pi}{\lambda_G} \cdot \Phi(x, y), \quad (\text{S35})$$

$$\varphi(x, y, \lambda_B) = \frac{2\pi}{\lambda_B} \cdot \Phi(x, y), \quad (\text{S36})$$

where λ_R , λ_G and λ_B are the wavelengths for Red, Green and Blue colors, respectively. By incorporating the phase profiles in Eqs. S34-S36 into metasurfaces, we can get aberration-free log-polar coordinate transformation at λ_R , λ_G and λ_B .

Figure S19. The design of meta-atoms for the RGB log-polar coordinate transformation metasurfaces. (a) The schematic of a meta-atom consisting of one silicon nanobar for Red, one silicon nanobar for Green, and two silicon nanobars for Blue. (b) The wavelength dependent right-to-left circular polarization conversion efficiency. Blue curve: only the Blue nanobars are enabled, Green curve: only the green nanobar is enabled; Red curve: only the Red nanobar is enabled; Black curve: all the four nanobars are enabled. (c) The PB phase as a function of rotation angle of the nanobars for $\lambda_R=636$ nm, $\lambda_G=540$ nm and $\lambda_B=465$ nm. The curve for the ideal PB phase is used for reference.

To verify our theoretical prediction, we designed a log-polar coordinate transformation metasurface operating for Red, Green, and Blue colors (Fig. S19). We utilized crystalline silicon to build metasurfaces, whose absorption in visible spectrum is much smaller than amorphous silicon. To achieve image transformation for RGB colors, each meta-atom consists of four silicon nanobars with a height of 500 nm. For the Red nanobar in Fig. S19a, the length $l_r=140$ nm and the width $w_r=110$ nm, which gives a maximum right-to-left circular polarization conversion efficiency at $\lambda_R=636$ nm. Similarly, for the

Green nanobar in Fig. S19a, the length $l_g=110$ nm and the width $w_g=40$ nm, which yields a peak right-to-left circular polarization conversion efficiency at $\lambda_G=540$ nm. Due to the weak scattering efficiency at short wavelength, we arranged two nanobars for the phase control of blue light, and each Blue nanobar has a length of $l_b=80$ nm and a width of $w_b=40$ nm. The maximum right-to-left circular polarization conversion efficiency for the blue nanobars is at $\lambda_B=465$ nm.

We then calculated the polarization conversion efficiency for the meta-atom consisting of those four nanobars. Compared with the efficiency for the meta-atoms with single color nanobars, the spectrum peaks are almost the same indicating weak cross-talk between nanobars. On the other hand, we utilized Pancharatnam-Berry (PB) phase method to control the phase imparted on the incident light. We calculated the imparted phase delay as a function of the rotation angle of nanobars. As can be seen in Fig. 19c, the phase delays for all the three wavelengths match the ideal PB phase very well.

To further evaluate the cross-talk between nanobars, we calculated the energy distribution for the meta-atom at $\lambda_B=465$ nm, $\lambda_G=540$ nm and $\lambda_R=636$ nm. As shown in Fig. S20, for each wavelength incidence, only the correlated nanobars have light field tightly confined with barely cross-talk between nanobars.

Figure S20. The spatial distribution of a meta-atom for $|E|^2$ at $\lambda_B=465$ nm, $\lambda_G=540$ nm and $\lambda_R=636$ nm. When the incident plane wave right circular polarized light wavelength is $\lambda_B=465$ nm, $\lambda_G=540$ nm and $\lambda_R=636$ nm, only the Blue, Green and Red nanobars are activated, respectively.

By using the nanobars in Fig. S19, we constructed a metasurface operating at $\lambda_B=465$ nm, $\lambda_G=540$ nm and $\lambda_R=636$ nm, and performed log-polar coordinate transformation for the ring-shaped image in Fig. 3a in the main text. Unlike the significant chromatic aberration for the metasurface designed with a single nanobar in Fig. S18, the transformed images using the RGB metasurface are almost the same for $\lambda_B=465$ nm, $\lambda_G=540$ nm and $\lambda_R=636$ nm (Fig. S21). Therefore, our metasurfaces can operate in the visible wavelength regime with at three RGB wavelengths.

Figure S21. The log-polar coordinate transformed images using metasurfaces at different wavelengths in the visible spectrum.

”

We have also included the citation of the newly added Supplementary Note 10 in the main text (see below):

“

... ...Furthermore, although our metasurfaces can only work for the log-polar coordinate transformation of binary images, it is also possible to transform color images by using metasurfaces operating at red, green, and blue wavelengths (Supplementary Note 10)²².

”

1.9. Overall writing should be improved. There are several grammar issues (e.g, Page 8, Line 143-145) and different fonts (Page 8, Line 193-196).

We appreciate the reviewer's attention to detail. The italicized font was initially intended for emphasis. We have now adjusted them to standard font in the revised manuscript. We have also corrected the grammar issues.

The second paragraph on page 8 in the revised main text (previously Line 143-145) now reads "... ..transformations. We initially demonstrated the geometric transformation for binary images (i.e., images whose pixels have two possible intensity values: 0 and 1) and realized Cartesian to log-polar coordinate transformation as an example. In this instance, the metasurface"

The first paragraph on page 11 in the revised main text (previously Line 193-196) now reads "... ..converted into translations in the new coordinate system ($X' = X + a \cdot \ln s$, $Y' = Y - a \cdot \alpha$, Supplementary Note 8). According to the shift theorem of the correlation function $R_{g_1 g_i}(X + X_0, Y + Y_0) = \mathcal{F}^{-1}\{[\mathcal{F}(g_1(X, Y))]^* \mathcal{F}(g_i(X + X_0, Y + Y_0))\}$, the translation of g_i can shift the correlation $R_{g_1 g_i}$ with the same amount without deforming the appearance of $R_{g_1 g_i}$. Therefore, by performing 2D correlation analysis, we can not only accurately recognize the test images, but also quantitatively determine their scale factors and rotation angles."

Responses to Reviewer #2:

The manuscript by Zhang et al. proposes a metasurface-based all-optical geometric image transformation that plays a vital role in various fields of image processing, such as pattern recognition, target tracking, and image registration. Specifically, with the aid of the designed metasurface, the transformed input image into the spatial frequency domain is invariant to scaling and rotation, which is highly useful for the image classification. Compared to conventional spatial light modulator or diffractive optical element-based optical analog image transformations, the proposed metasurface suppresses undesirable high-order diffraction orders, and increases spatial resolution within a miniaturized form factor. In addition, the authors have specially engineered meta-atoms to ensure that output light carries the designed phase by optimizing the geometric parameters of meta-atoms which minimize the neighboring coupling effects between adjacent meta-atoms. The authors provide a clear and thorough explanation of the simulations and the experimental results. The paper is timely and should be of great interest to a broad range of communities including optics, computing, and information processing. Therefore, I believe this manuscript deserves publication in Nature Communications once some questions are properly addressed by the author.

We are deeply grateful to the reviewer for their constructive feedback and for recognizing the novelty of our work in image classification.

We also extend our gratitude for the reviewer's insightful comments, which have greatly enhanced our manuscript. In light of the feedback, we have strengthened our revised manuscript with further theoretical discussions and simulation outcomes, specifically in the newly added Supplementary Note 2 (in response to Comment #2.1), as well as additional design and simulation data for metasurfaces for multi-color image geometric transformations in Supplementary Information Notes 10 (in response to Comment #2.2).

In the following sections, we will address each of the reviewer's concerns in detail to provide a comprehensive understanding of our findings.

2.1. The importance of log-polar transformation in image classification is well explained. It is understood that the metasurface conducting log-polar transformation can work with various input images. I recommend the authors to comment on any conditions regarding the input images, such as their physical size, and how these conditions may affect the design of the metasurface.

We thank the reviewer for these insightful suggestions. In our work, we employed the small angle approximation to derive the spatial phase gradient for the geometric image transformation. This introduced limitations on the input field of view (FoV). Specifically, we considered $\sin \beta \approx \tan \beta = \frac{X(x,y)}{f}$ in the (x, z) plane and $\sin \beta \approx \tan \beta = \frac{Y(x,y)}{f}$ in the (y, z) plane. This approximation holds true primarily when β is small. To evaluate the precision for the small angle approximation, we undertook additional calculation to analyze the associated error. Furthermore, we conducted simulations to study the impact of small angle approximation on the geometric image transformation. Our findings indicate that for an excessively large image size (requires a large FoV), the transformed image may be distorted due to the error introduced by the approximation. Conversely, if the input image is too small, distortion can arise as the image exceeds the resolution of the optical system. It is important to note that in our experiment, the input image sizes were chosen to align with the appropriate FoV and resolution range of the system, ensuring that the distortion in the transformed image was negligible.

We have added the detailed analysis in the revised Supplementary as Note 2 (see below).

“

2.The small angle approximation

In Supplementary Note 1, we used small angle approximation to simplify the derivation and obtain $\sin \beta \approx \tan \beta = \frac{X(x,y)}{f}$ in the (x,z) plane and $\sin \beta \approx \tan \beta = \frac{Y(x,y)}{f}$ in the (y,z) plane. In this section, we evaluated the accuracy for the small angle approximation and the effect on the geometric image transformation.

Taking the log-polar coordinate transformation as an example. Combining Eqs. S5 and S10, we can derive $\sin \beta \approx \tan \beta = \frac{a}{f} \cdot \ln \frac{r}{b}$ in the (x,z) plane. Similarly, we have $\sin \beta \approx \tan \beta = \frac{-a \cdot \alpha}{f}$ in the (y,z) plane. In our work, $a = 30 \mu m$, $b = 100 \mu m$, $f = 400 \mu m$ and $\alpha \in (-\pi, \pi)$. In the (y,z) plane, we have $\tan \beta \in (-0.2356, 0.2356)$ and the angle range $\beta \in (-0.2314, 0.2314)$. For such an angle range, the range for $\sin \beta$ is $(-0.2293, 0.2293)$ which yields a maximum error of 0.92%. In the (x,z) plane, the angle range of β and the error are proportional to r (i.e. the size of image). The images we used in this work are all below $100 \mu m$, corresponding to an angle of 0.0137 rad and an error of 0.94%.

Figure S3. The log-polar coordinate transformation for different size of ring-shaped images. Upper: theoretical calculation results. Bottom: simulation results. r_o is the outer radius for the ring shapes, and the inner radius is fixed as a half of the outer radius.

To better understand the effect of image size on the geometric image

transformation, we simulated the log-polar transformation for different sizes of ring-shaped images and compared with theoretical prediction. As can be seen in Fig. S3, the transformed images have a relatively smaller distortion when the outer radius r_o of the ring shapes between $40 \mu m$ and $100 \mu m$. For the rings with $r_o > 100 \mu m$, due to the increased error for the small angle approximation, the transformed images are distorted. For the ring-shaped image with $r_o=20 \mu m$, the effective numerical aperture for the metasurfaces $NA = r_o/f = 0.05$, which yields a spatial resolution of $13 \mu m$ at the operation wavelength $\lambda = 1.064 \mu m$. Consequently, due to the spatial resolution limitation, the transformed image for $r_o=20 \mu m$ is also distorted. Therefore, our metasurfaces can only operate for the geometric transformation of images with the size in the range between $40 \mu m$ and $100 \mu m$ with negligible aberration.

”

We have also included the citation of the newly added Supplementary Note 2 in the main text, and copied the corresponding sentences below for the reviewer’s convenience.

The last paragraph on page 5 in the main text now reads:

“

... ..Under small angle approximation (Supplementary Note 2),¹⁸ $\sin \beta \approx \tan \beta = \frac{X(x,y)}{f}$. Similarly, in the (y,z) plane, we have $\sin \beta = \frac{1}{k} \cdot \frac{\partial \varphi_0(x,y)}{\partial y} \approx \tan \beta = \frac{Y(x,y)}{f}$. Therefore, we derived the phase gradient of $\varphi_0(x,y), \dots \dots$

”

2.2. For image classification, it seems that the metasurface operates in the visible wavelength regime with at least three wavelengths, RGB, being preferred. In addition to the discussion on scale- and rotation-invariant properties of the proposed metasurfaces, I recommend the authors to add a discussion on the wavelength-dependent variance and the resulting offset on the transformed plane.

We thank the reviewer for these insightful suggestions. We have incorporated discussions on the wavelength dependence and developed a multi-color metasurface that enables geometric transformations for RGB color images. Detailed discussions can be found in our response to Comment #1.8, which is also provided below for the reviewer’s convenience.

For images containing color information (i.e., wavelength information), we have conducted additional studies. **First**, we analyzed the phase profiles for log-polar coordinate transformation and found that the image would be distorted due to chromatic phase aberration when the operational wavelength changed. **Then**, we simulated the transformation of a ring-shaped image at different wavelengths. The results revealed that the transformed images became bent rectangles because of the chromatic aberration.

To overcome the chromatic aberration, we designed a new set meta-atoms that works at three distinct wavelengths (red, green, and blue). We incorporated the phase profile $\varphi(x, y)$ for each of these wavelengths separately to avoid the chromatic aberration. **Finally**, we performed the log-polar coordinate transformation for the same ring-shaped image. Unlike the significant chromatic aberration observed with previous metasurface design, the transformed images using the new RGB metasurface remained consistent across red, green, and blue colors. Therefore, we have demonstrated that our metasurfaces can operate for RGB color images.

We have now added the corresponding contents in the revised Supplementary Note 10, and also included below.

“

10. Metasurfaces for multi-color geometric image transformations

From Eq. S15, we know the phase profile $\varphi(x, y)$ of metasurfaces for log-polar coordinate transformation is wavelength dependent. Therefore, we can re-write it as

$$\varphi(x, y, \lambda) = \frac{2\pi}{\lambda} \cdot \left\{ \frac{1}{f} \cdot [x \cdot X(x, y) + y \cdot Y(x, y) - a \cdot x] - \sqrt{x^2 + y^2 + f^2} \right\}. \quad (\text{S31})$$

For simplicity, we can obtain

$$\varphi(x, y, \lambda) = \frac{2\pi}{\lambda} \cdot \Phi(x, y), \quad (\text{S32})$$

$$\Phi(x, y) = \frac{1}{f} \cdot [x \cdot X(x, y) + y \cdot Y(x, y) - a \cdot x] - \sqrt{x^2 + y^2 + f^2}, \quad (\text{S33})$$

where $\Phi(x, y)$ is wavelength independent. In our work, the metasurfaces were design at $\lambda = 1064 \text{ nm}$. Therefore, if we change the wavelength of the incident laser, the transformed image will be distorted as a result of the chromatic phase aberration (Eq. S32).

On the other hand, the transmittance (i.e. LCP-to-RCP conversion efficiency) for our meta-atom was also wavelength dependent. As shown in Fig. S17, our meta-atom has a peak transmittance of about 98% at $\lambda=1064 \text{ nm}$. When the wavelength deviates from $\lambda=1064 \text{ nm}$, the transmittance drops down to about 60% at $\lambda=980 \text{ nm}$ and $\lambda=1200 \text{ nm}$. Therefore, if we change the wavelength of the incident laser, the transformed image will become dim due to smaller transmittance efficiency.

Figure S17. LCP-to-RCP conversion efficiency for our meta-atoms at different wavelengths.

To further study the wavelength dependent characteristics for our log-polar coordinate transformation, we simulated the transformation for the ring-shaped

image in Fig. 3a in the main text at different wavelengths, and the results are plotted in Fig. S18. When the wavelength deviates from $\lambda=1064$ nm, the transformed images turn dim due to smaller transmittance. In addition, when the wavelength is beyond ± 20 nm away from $\lambda=1064$ nm, the transformed images become bended rectangles due to the chromatic aberration (Eq. S32)

Figure S18. The log-polar coordinate transformed images using a metasurface at different wavelengths

To overcome the chromatic aberration, we need to incorporate the wavelength dependence characteristics for the phase profile in Eq. S32. To this end, we can design the meta-atoms working at three different wavelengths (e.g. Red, Green and Blue), and design the phase profile $\varphi(x, y)$ for the three different wavelengths separately to avoid the chromatic aberration. As such we can obtain

$$\varphi(x, y, \lambda_R) = \frac{2\pi}{\lambda_R} \cdot \Phi(x, y), \quad (\text{S34})$$

$$\varphi(x, y, \lambda_G) = \frac{2\pi}{\lambda_G} \cdot \Phi(x, y), \quad (\text{S35})$$

$$\varphi(x, y, \lambda_B) = \frac{2\pi}{\lambda_B} \cdot \Phi(x, y), \quad (\text{S36})$$

where λ_R , λ_G and λ_B are the wavelengths for Red, Green and Blue colors, respectively. By incorporating the phase profiles in Eqs. S34-S36 into metasurfaces, we can get aberration-free log-polar coordinate transformation at λ_R , λ_G and λ_B .

Figure S19. The design of meta-atoms for RGB log-polar coordinate transformation metasurfaces. (a) The schematic of a meta-atom consisting of one silicon nanobar for Red, one silicon nanobar for Green, and two silicon nanobars for Blue. (b) The wavelength dependent right-to-left circular polarization conversion efficiency. Blue curve: only the Blue nanobars are enabled, Green curve: only the green nanobar is enabled; Red curve: only the Red nanobar is enabled; Black curve: all the four nanobars are enabled. (c) The PB phase as a function of rotation angle of the nanobars for $\lambda_R=636$ nm, $\lambda_G=540$ nm and $\lambda_B=465$ nm. The curve for the ideal PB phase is used for reference.

To verify our theoretical prediction, we designed a log-polar coordinate transformation metasurface operating for Red, Green and Blue colors (Fig. S19). We utilized crystalline silicon to build metasurfaces, whose absorption in visible spectrum is much smaller than amorphous silicon. To achieve image transformation for RGB colors, each meta-atom consists of four silicon nanobars with a height of 500 nm. For the Red nanobar in Fig. S19a, the length $l_r=140$ nm and the width $w_r=110$ nm, which gives a maximum right-to-left

circular polarization conversion efficiency at $\lambda_R=636$ nm. Similarly, for the Green nanobar in Fig. S19a, the length $l_g=110$ nm and the width $w_g=40$ nm, which yields a peak right-to-left circular polarization conversion efficiency at $\lambda_G=540$ nm. Due to the weak scattering efficiency at short wavelength, we arranged two nanobars for the phase control of blue light, and each Blue nanobar has a length of $l_b=80$ nm and a width of $w_g=40$ nm. The maximum right-to-left circular polarization conversion efficiency for the blue nanobars is at $\lambda_B=465$ nm.

We then calculated the polarization conversion efficiency for the meta-atom consisting of those four nanobars. Compared with the efficiency for the meta-atoms with single color nanobars, the spectrum peaks are almost the same indicating weak cross-talk between nanobars. On the other hand, we utilized Pancharatnam-Berry (PB) phase method to control the phase imparted on the incident light. We calculated the imparted phase delay as a function of the rotation angle of nanobars. As can be seen in Fig. 19c, the phase delays for all the three wavelengths match the ideal PB phase very well.

To further evaluate the cross-talk between nanobars, we calculated the energy distribution for the meta-atom at $\lambda_B=465$ nm, $\lambda_G=540$ nm and $\lambda_R=636$ nm. As shown in Fig. S20, for each wavelength incidence, only the correlated nanobars have light field tightly confined with barely cross-talk between nanobars.

Figure S20. The spatial distribution of a meta-atom for $|E|^2$ at $\lambda_B=465$ nm, $\lambda_G=540$ nm and $\lambda_R=636$ nm. When the incident plane wave right circular polarized light wavelength is $\lambda_B=465$ nm, $\lambda_G=540$ nm and $\lambda_R=636$ nm, only

the Blue, Green and Red nanobars are activated, respectively.

By using the nanobars in Fig. S19, we constructed a metasurface operating at $\lambda_B=465$ nm, $\lambda_G=540$ nm and $\lambda_R=636$ nm, and performed log-polar coordinate transformation for the ring-shaped image in Fig. 3a in the main text. Unlike the significant chromatic aberration for the metasurface designed with a single nanobar in Fig. S18, the transformed images using the RGB metasurface are almost the same for $\lambda_B=465$ nm, $\lambda_G=540$ nm and $\lambda_R=636$ nm (Fig. S21). Therefore, our metasurfaces can operate in the visible wavelength regime with at three RGB wavelengths.

Figure S21. The log-polar coordinate transformed images a metasurface at different wavelengths in the visible spectrum.

”

We have also included the citation of the newly added Supplementary Note 10 in the main text (see below):

“

... ..Furthermore, although our metasurfaces can only work for the log-polar coordinate transformation of binary images, it is also possible to transform color images by using metasurfaces operating at red, green, and blue wavelengths (Supplementary Note 10)²².

”

2.3. In Figure S3, the effect of neighboring coupling between adjacent meta-atoms is discussed by comparing analytical and FDTD results. However, this analysis was conducted on nanodisks. I recommend the authors to compare star and diamond rectangular meta-atoms, as denoted in Fig. 2C of the main manuscript, using FDTD simulations to emphasize the effectiveness of the specially engineered meta-atoms.

We thank the reviewer for the comments. **We did compare the rectangular meta-atoms denoted by the star and the diamond, as denoted in Fig. 2C of the main manuscript, using FDTD simulations in Figure S6 (i.e., Figure S8 in the revised Supplementary Information).**

In our work, we first evaluated the performance of nanodisks based metasurfaces for geometric image transformation. We found that the nanodisks based metasurfaces have serious neighboring coupling effect and low fabrication tolerance, both of which can severely degrade the performance of geometric image transformation. The corresponding text along with analytical, simulation and experimental data was included in the revised Supplementary Note 3.

Then we utilized nano-bars to convert incident circular polarized light to its orthogonally polarized counterpart with specific abrupt phase shifts that are linearly dependent on the nano-bars' orientation angles. Although the neighboring coupling for nano-bars are smaller than that of nanodisks, the actual phase can still deviate from ideal geometric phases with a phase variance (σ^2). **To this end, we took two steps to minimize the neighboring coupling between adjacent meta-atoms.**

Firstly, we calculated the LCP-to-RCP conversion efficiency and the phase variance (σ^2) for nano-bars with various sizes in Fig. 2C of the main text. The star and diamond designs in Fig. 2C represent two meta-atoms with similar η but different σ^2 , located at (330 nm, 140 nm) and (360 nm, 170 nm), respectively.

We also did FDTD simulations to compare the geometric image transformation for both the star and diamond designs in Fig. 2C, and the results was included in the revised Supplementary Note 4. The star design in Fig. 2C with smaller phase variance (σ^2) displayed much better performance than that of diamond design.

Secondly, although the star design has smaller phase variance, it is still deviate from ideal geometric phases (Fig. 2d). That is to say, the geometric phases follow the rotation angle of the meta-atom in a nonlinear manner. We also compared the geometric image transformation performance with the following two meta-atom designs: 1# the star design meta-atom following the nonlinear dependence of phase on orientation angle as shown in Fig. 2d in the main text; 2# the star design meta-atom following the linear dependence of phase on orientation angle. The FDTD simulation results in the Supplementary Note 4 indicates that the 1# method has better performance than that of the 2# method.

We have also provided the relevant sections below for the reviewer's convenience.

“

4. The optimization for metasurface design

To evaluate the performance of our optimized metasurface design in Fig. 2 in the main text, we first used our selected meta-atom ($l_x=330$ nm, $l_y=140$ nm, labeled by a star in Fig. 2c in the main text) to construct a metasurface for Gaussian-to-flat-top transformation, and the spatial arrangement of meta-atoms followed the linear dependence of phase on orientation angle. As can be seen in Fig. S8a, the output beam profile is close to a flat-top shape, indicating much less neighboring coupling effects than that of the nanodisk meta-atoms in Fig. S7d. We also calculated the output flat-top beam profile using a meta-atom with $l_x=350$ nm and $l_y=170$ nm (labeled by a diamond in Fig. 2c in the main text) as a comparison. This meta-atom has the same conversion efficiency (η) as the selected one but with a larger phase variance (σ^2) (Fig. 2c in main

text). Consequently, the output flat-top beam profile for this metasurface is severely distorted as shown in Fig. S8b. Therefore, our selected meta-atom with $l_x=330$ nm and $l_y=140$ nm has the minimized neighboring coupling effects. To further reduce the effects of neighboring coupling, we spatially arranged meta-atoms in the metasurface by following the nonlinear dependence of phase on orientation angles as shown in Fig. 2d in the main text. Figure S8c reveals that the output flat-top beam profile for the meta-atoms arranged by nonlinear phase dependence exhibits a better flatness than that in Fig. S8a, indicating weaker neighboring coupling effects.

Figure S8. Evaluation of the performance of optimized metasurfaces. (a) The transformed flat-top beam profile by using the selected meta-atom ($l_x=330$ nm, $l_y=140$ nm, labeled by a star in Fig. 2c in the main text) with the meta-atom spatial arrangement following the linear dependence of phase on orientation angle. (b) The transformed flat-top beam profile by metasurfaces by using a meta-atom ($l_x=360$ nm, $l_y=170$ nm, labeled by a diamond in Fig. 2c in the main text) which has the same conversion efficiency as the selected meta-atom ($l_x=330$ nm, $l_y=140$ nm) but higher neighboring coupling effects. (c) The transformed flat-top beam profile by using the selected meta-atom ($l_x=330$ nm, $l_y=140$ nm) with the meta-atom spatial arrangement following the nonlinear dependence of phase on orientation angle as shown in Fig. 2d in the main text to further minimize the neighboring coupling effects.

”

Responses to Reviewer #3:

The authors report optical image transformation using all-dielectric metasurface. They intend to produce large changes in optical images; one of the examples is Cartesian to log-polar coordinate transformation.

The design principle is similar to metalens and holography using metasurfaces, adjusting phase of light finely.

As far as I examined this manuscript (MS), the most practical application is to produce flat top beam (Fig. 5).

To consider this MS further, the following points should be clarified.

We thank the reviewer for these comments. As a result of these insightful comments, we believe that our revised manuscript is much stronger now with the additional analysis on small angle approximation in the newly added Supplementary Note 2 (response to Comment #3.2), the additional data on the sample fabrication and experimental setup in the newly added Supplementary Note 5 and the revised Supplementary Note 6 (response to Comment #3.3), the additional data on the log-polar to Cartesian coordinate transformation in the newly added Supplementary Note 9 (response to Comment #3.4 and #3.5).

We will respond below to the reviewer's concerns point-by-point to clarify our interpretation of our results. However, first we would like to emphasize the significance of our work on image geometric transformation and the fundamental differences of our work from metalens and holography using metasurfaces.

For metalens and holography using metasurfaces, the phase profiles are designed by assuming the entire metasurface is illuminated by light, and the outputs are solely determined by the phase profiles of metasurfaces. Therefore, once the metasurface is fabricated, the metasurface must be altered by post-fabrication methods to form new phase profiles to

produce other holographic images.

For our geometric image transformation using metasurfaces, the input images are projected onto the metasurfaces using a 4-f system, and the metasurface acts as a “computer”, which establishes a connection between the input images and the output images by the coordinate transformation relationships. Therefore, our metasurfaces can be used to calculate and output the transformed images for any image projected onto the metasurfaces. Each input image has its own transformed image which is determined by the coordinate transformation relationship formed by the metasurface. Therefore, to produce new output images, we only need to change the input image projected on the metasurface without modifying the metasurface.

We thank the reviewer for pointing out the practical application of our work for flat-top beam transformation. In addition to the ultra-high spatial resolution and high-order-diffraction-free property endowed by the subwavelength-scale meta-atoms in our metasurfaces, we also want to stress that the planar nature of our metasurfaces allows for the potential vertical integration of multiple metasurfaces, aiming at even more sophisticated optical functionalities.

To verify the vertical integration capability of our metasurfaces for new functionalities, we conducted additional experiments. As can be seen in Fig. R1 below, we integrated another layer of metasurface on the opposite side of the fused silica. The front layer of the metasurface is utilized to transform the geometric shape and intensity distribution, while the back layer of the metasurface is used to compensate the phase, such that the final output beam is expected to have a square-shaped flat-top intensity profile with uniform phase distribution for collimated propagation. It should be noted that collimated flat-top beams are of importance to microscopy illumination and MOPA lasers, where long working ranges are required.

Bearing this in mind, we designed a doublet metasurface to transform a Gaussian beam with a radius of $r_0=200\ \mu\text{m}$ to a square flat-top beam with a

width of $2w_0=400\ \mu\text{m}$ as a proof-of-concept demonstration. In order to present a clear comparison, we first studied the beam evolution with the front metasurface only. As can be seen in Fig. R2a and 2b, the input Gaussian beam gradually evolved into a flat-top beam at the back surface of the fused silica chip (*i.e.*, $z=1\ \text{mm}$). Without phase correction, the flat-top beam diverged quickly after passing through the back side of the fused silica chip with a deterioration of the beam uniformity (Fig. R2a). In our experiment, as shown in Figs. R2d and R2e, we also observed that the beam exhibited a quick divergence behavior.

To achieve collimated flat-top beam propagation, we added another layer of metasurface on the back side of the chip for phase correction. We calculated the flat-top beam evolution after passing through the second layer metasurface. In contrast to the quick divergence shown in Fig. R2a, the beam for the doublet maintained uniform for 2 mm propagation (Fig. R2c). We also fabricated the second layer metasurface in experiment for phase correction. As can be seen in Fig. R2f, we observed the collimated flat-top beam was collimated as long as 2 mm propagation.

Figure R1. Integrated metasurface doublet for collimated Gaussian-to-flat-top laser beam transformation. (a) The schematic for the integrated metasurface doublet. Two metasurfaces are fabricated on both sides of a fused silica with a thickness of 1mm. (b) The phase profile for Metasurface #1. (c) The phase profile for Metasurface #2.

Figure R2. Demonstration of collimated Gaussian-to-flat-top beam transformation using an integrated metasurface doublet. The simulated (a) and measured (d) beam evolution from the first layer metasurface ($z=0$ mm) to $z=3$ mm without the second layer of metasurfaces. The dashed white lines indicate the back surface of the fused silica chip. The simulated (b) and measured (c) intensity profiles for the input Gaussian beam (top) and the output flat-top beam (bottom). The simulated (c) and measured (f) beam evolution from the second layer of metasurface ($z=1$ mm) to $z=3$ mm for the metasurface doublet.

3.1 Definitions are unclear.

Lines 79,80: k , f are not defined explicitly.

Line 93: θ is defined as deflection angle. However, θ is assigned to rotation angle in line 125.

We thank the reviewer for pointing out these errors. We have now made them clear in the revised manuscript. The first paragraph on page 5 (including previously lines 79, 80, and 93) now reads

“... .. the phase of a Fourier transform lens $\varphi_f(x, y) = -k \cdot \sqrt{x^2 + y^2 + f^2}$,

where k is the wave number and f is the focal length.... ..for normal light incidence on the metasurface with a phase profile of $\varphi_0(x, y)$, the local light deflection angle in the (x, z) plane can be expressed as $\sin \beta = \frac{1}{k} \cdot \frac{\partial \varphi_0(x, y)}{\partial x}$. The light is subsequently modulated by the Fourier transform phase profile $\varphi_f(x, y)$ and mapped onto the spatial frequency domain (X, Y) . Under small angle approximation (Supplementary Note 2),¹⁸ $\sin \beta \approx \tan \beta = \frac{X(x, y)}{f}$. Similarly, in the (y, z) plane, we have $\sin \beta = \frac{1}{k} \cdot \frac{\partial \varphi_0(x, y)}{\partial y} \approx \tan \beta = \frac{Y(x, y)}{f}$. Therefore, we derived the phase gradient of $\varphi_0(x, y), \dots$..”

3.2 Small angle approximation

Line 95: Small angle approximation is used. But there is no justification.

Actual range of theta should be noted.

We thank the reviewer for the suggestion. In our work, we employed the small angle approximation to derive the spatial phase gradient for the geometric image transformation. This introduced limitations on the input field of view (FoV). Specifically, we considered $\sin \beta \approx \tan \beta = \frac{X(x, y)}{f}$ in the (x, z) plane and $\sin \beta \approx \tan \beta = \frac{Y(x, y)}{f}$ in the (y, z) plane. This approximation holds true primarily when β is small. To evaluate the precision for the small angle approximation, we undertook additional calculation to analyze the associated error. Furthermore, we conducted simulations to study the impact of small angle approximation on the geometric image transformation. Our findings indicate that for an excessively large image size (requires a large FoV), the transformed image may be distorted due to the error introduced by the approximation. Conversely, if the input image is too small, distortion can arise as the image exceeds the resolution of the optical system. **It is important to note that in our experiment, the images we used in this work are all below 100 μm , corresponding to an angular range of 0.0137 rad, which yields a maximum**

error of 0.94%. The input image sizes were chosen to align with the appropriate FoV and resolution range of the system, ensuring that the distortion in the transformed image was negligible.

We have added the detailed analysis in the revised Supplementary as Note 2 (see below).

“

2.The small angle approximation

In Supplementary Note 1, we used small angle approximation to simplify the derivation and obtain $\sin \beta \approx \tan \beta = \frac{X(x,y)}{f}$ in the (x, z) plane and $\sin \beta \approx \tan \beta = \frac{Y(x,y)}{f}$ in the (y, z) plane. In this section, we evaluated the accuracy for the small angle approximation and the effect on the geometric image transformation.

Taking the log-polar coordinate transformation as an example. Combing Eqs. S5 and S10, we can derive $\sin \beta \approx \tan \beta = \frac{a}{f} \cdot \ln \frac{r}{b}$ in the (x, z) plane. Similarly, we have $\sin \beta \approx \tan \beta = \frac{-a \cdot \alpha}{f}$ in the (y, z) plane. In our work, $a = 30 \mu m$, $b = 100 \mu m$, $f = 400 \mu m$ and $\alpha \in (-\pi, \pi)$. In the (y, z) plane, we have $\tan \beta \in (-0.2356, 0.2356)$ and the angle range $\beta \in (-0.2314, 0.2314)$. For such an angle range, the range for $\sin \beta$ is $(-0.2293, 0.2293)$ which yields a maximum error of 0.92%. In the (x, z) plane, the angle range of β and the error are proportional to r (i.e. the size of image). The images we used in this work are all below $100 \mu m$, corresponding to an angle of 0.0137 rad and an error of 0.94%.

Figure S3. The log-polar coordinate transformation for different size of ring-shaped images. Upper: theoretical calculation results. Bottom: simulation results. r_o is the outer radius for the ring shapes, and the inner radius is fixed as a half of the outer radius.

To better understand the effect of image size on the geometric image transformation, we simulated the log-polar transformation for different sizes of ring-shaped images and compared with theoretical prediction. As can be seen in Fig. S3, the transformed images have a relatively smaller distortion when the outer radius r_o of the ring shapes between $40 \mu m$ and $100 \mu m$. For the rings with $r_o > 100 \mu m$, due to the increased error for the small angle approximation, the transformed images are distorted. For the ring-shaped image with $r_o = 20 \mu m$, the effective numerical aperture for the metasurfaces $NA = r_o/f = 0.05$, which yields a spatial resolution of $13 \mu m$ at the operation wavelength $\lambda = 1.064 \mu m$. Consequently, due to the spatial resolution limitation, the transformed image for $r_o = 20 \mu m$ is also distorted. Therefore, our metasurfaces can only operate for the geometric transformation of images with the size in the range between $40 \mu m$ and $100 \mu m$ with negligible aberration.

”

We have also included the citation of the newly added Supplementary Note 2 in the main text. The last paragraph on page 5 in the main text now reads:

“

... ..Under small angle approximation (Supplementary Note 2),¹⁸ $\sin \beta \approx \tan \beta = \frac{X(x,y)}{f}$. Similarly, in the (y,z) plane, we have $\sin \beta = \frac{1}{k} \cdot \frac{\partial \varphi_0(x,y)}{\partial y} \approx \tan \beta = \frac{Y(x,y)}{f}$. Therefore, we derived the phase gradient of $\varphi_0(x,y), \dots \dots$

”

3.3 Fig. 3, 4

Actual optical and SEM images of the metasurfaces are highly preferred to be shown.

In Fig. 3, how was the break of the ring implemented in the metasurface structure?

We thank the reviewer for this excellent suggestion. In response, we have added the corresponding text along with both the optical and SEM images of our samples in the revised Supplementary as Note 5.

The ring-shaped image was realized by an aluminum-coated glass slide, which was patterned to create a ring-shaped transparent window. **By illuminating the patterned aluminum-coated glass slide, the transparent region rendered a ring-shaped image. This image was subsequently projected onto the metasurface using a 4-f system.** We have added the corresponding text along with the schematic of the optical setup in the revised Supplementary Note 6.

We have included the relevant sections below for the reviewer’s convenience. Supplementary Note 5 now reads:

“

5. Sample fabrication

To fabricate metasurfaces for geometric image transformation, we first deposited a layer of amorphous silicon film with a thickness of 500 nm on a fused silica substrate by Plasma Enhanced Chemical Vapor Deposition (PECVD). Then we spin-coated e-beam resist on top of the silicon film followed

by e-beam lithography. After development, we deposited a thin layer of Aluminum on the sample as the hard mask by e-beam evaporation for the dry etching followed by a lift-off process. Then we transferred the pattern onto the silicon layer by using Inductively Coupled Plasma - Reactive Ion Etching (ICP-RIE). Finally, we removed the residual Aluminum layer by wet-etching method. Figure S9 shows the optical microscopy image and the scanning electron microscopy (SEM) image for a representative fabricated metasurface for geometric image transformation.

Figure S9. Optical microscopy image (a) and scanning electron microscopy (SEM) image (b) for a representative fabricated metasurface for geometric image transformation.

To fabricate the test images for log-polar coordinate transformation, we first deposited a layer of Aluminum film with a thickness of 100 nm on a glass slide by e-beam evaporation. The transmittance for 100 nm-thick Aluminum film is about 10^{-6} around $\lambda = 1064$ nm, which is small enough to block the incident laser beam. Then we spin-coated a layer of photo-resist on top of Aluminum film. After that, we used a laser writer to expose the patterns of the test images on the photo-resist. After development, we transferred the pattern onto the

Aluminum film by using Inductively Coupled Plasma - Reactive Ion Etching (ICP-RIE). As such, the transparent region of the patterned Aluminum film formed the test images for log-polar coordinate transformation (Fig. S10).

Figure S10. Optical microscopy images for the test images fabricated by a patterned Aluminum coated glass slide. The bright regions are transparent glass and the dark regions are 100nm-thick Aluminum coated glass. (a) A ring-shaped image; (b) A triangle-shaped image; (c) An airplane-shaped image.

”

The Supplementary Note 6 now reads:

“

6. Experimental setup

Figure S11. Schematic of the optical setup used for characterization. (a) The experiment setup for binary image transformation. (b) The experiment

setup for grayscale image transformation.

In our experiment, we utilized a collimated laser beam ($\lambda=1064$ nm) as the light source for illumination and used a linear polarizer and a quarter wave plate to prepare a circular polarization state to meet the requirement of the metasurface. We then used a 4× objective (NA=0.2) and a tube lens to project the transformed images on a camera.

For the binary image transformation, the collimated laser illuminated the patterned Aluminum coated glass slide to prepared the test images which were scaled down by 4× by a 4-f system and projected on the metasurface plane (Fig. S11a, Supplementary Note 3).

For the grayscale image transformation, the laser was coupled by a single-mode fiber to ensure single-mode Gaussian beam output. The Gaussian beam was then expanded by a beam expander and projected on the metasurface plane (Fig. S11b).

”

We have also revised the relevant description of sample fabrication and experimental characterization in the main text, and copied the corresponding sentences below for the reviewer’s convenience.

The third paragraph on page 9 in the main text now reads:

“

To experimentally demonstrate log-polar transformation using metasurfaces, we fabricated a metasurface incorporating the geometric transformation encoded phase $\varphi_0(x,y)$ and Fourier transform phase $\varphi_f(x,y)$ (**Supplementary Note 5**). **To prepare binary images, we patterned a piece of Aluminum coated glass, which can only allow the light to pass through the transparent region, forming binary images with only two intensity values.** To experimentally map an image from Cartesian to log-polar coordinate, we used a 4f system to project **the binary image** onto our nanofabricated metasurface, and the output transformed image was acquired by a camera

through an objective and a tube lens (Supplementary Note 6). In our experiment, for the projection of a ring-shaped image on the metasurface, we observed a rectangle image on the camera, which perfectly matched our prediction from simulation (Fig. 3b).

”

The last paragraph on page 13 in the main text now reads:

“

To experimentally realize the Gaussian-to-flat-top transformation with a metasurface, we fabricated a metasurface with the required phase profile encoded (Supplementary Note 5). To characterize the performance of the fabricated metasurface, we prepared the Gaussian grayscale image with a Gaussian laser beam with a waist diameter of 200 μm (Supplementary Note 6), and the metasurface was placed on the beam waist to ensure normal incidence (Fig. 6b).

”

3.4 Fig. 4

Is it possible to reproduce the original plane image based on Fig. 4d,g,j

(or e,h,k or f,i,l)?

If yes, how?

If no, why?

We thank the reviewer for the comment. Yes, it is possible to reproduce the original images through a log-polar to Cartesian coordinate transformation. We have conducted further analysis and derived the required phase profile for this transformation using a metasurface. For a detailed explanation, please see our response to Comment 3.5.

3.5 Log-polar to Cartesian

It is informative for readers to see optical transformation from log-polar to

Cartesian coordinate. Could the authors show the example(s)?

At least, the procedure(s) to realize them is expected to be described.

We thank the reviewer for these excellent suggestions. It is possible for a metasurface to perform log-polar to Cartesian coordinate transform of an image. We have further analyzed and identified the required phase profile for this transformation. By applying this, we transformed a rectangle – previously converted from a Cartesian coordinate ring – back into its original shape. Similarly, images from Fig. 4e were reverted to the airplane-shaped image in Fig. 4b. Further details are available in the revised Supplementary Note 9, which is also provided below for the reviewer's convenience.

“

9. Log-polar to Cartesian coordinate transformation using metasurfaces

In the main text, we have realized the image transformation from Cartesian to log-polar coordinate using metasurfaces. We can also utilize metasurfaces to perform log-polar to Cartesian coordinate transform of an image. In this case, we have a log-polar coordinate in the (x, y) plane:

$$x = a \cdot \ln \frac{r}{b}, \quad (\text{S23})$$

$$y = -a \cdot \alpha. \quad (\text{S24})$$

In the (X, Y) plane, we have the coordinate relations as below:

$$X(x, y) = r \cdot \cos(\alpha), \quad (\text{S25})$$

$$Y(x, y) = r \cdot \sin(\alpha), \quad (\text{S26})$$

$$r = b \cdot e^{\frac{x}{a}}, \quad (\text{S27})$$

$$\alpha = -\frac{y}{a}, \quad (\text{S28})$$

In order to obtain the encoded phase term $\varphi_0(x, y)$ of metasurfaces for the log-polar to Cartesian coordinate transform, we substituted the coordinate

transformation relations Eqs. S25-S28 into Eqs. S7-S8. By integrating the spatial phase gradient of $\varphi_0(x, y)$, we can then derive

$$\varphi_0(x, y) = \frac{k}{f} \cdot a \cdot X(x, y). \quad (\text{S29})$$

Therefore, combining Eq. S1, Eq. S2 and Eq. S29, we can obtain the phase profile $\varphi(x, y)$ of metasurfaces for log-polar to Cartesian coordinate transformation:

$$\varphi(x, y) = \frac{k}{f} \cdot a \cdot X(x, y) - k \cdot \sqrt{x^2 + y^2 + f^2}. \quad (\text{S30})$$

For the log-polar to Cartesian coordinate transformation, we considered only the geometric characteristics of the image $f(x, y)$ and neglected the grayscale information. Therefore, the metasurfaces for log-polar to Cartesian coordinate transformation only works for binary images whose pixels have only two possible intensity values (Eq. S9).

As we have demonstrated in Fig. 3 in the main text, the Cartesian to log-polar coordinate transformation for a ring is a rectangle. Conversely, for a rectangle in the log-polar coordinate, the transformed image in the Cartesian coordinate should be a ring. With this in mind, we projected a rectangle-shaped binary image as shown in Fig. S15a on a metasurface which was encoded by the phase profile in Eq. S30 to perform log-polar to Cartesian coordinate transformation. The rectangle image in Fig. S15a has the same size and location as the one in Fig. 3b in the main text, which was previously transformed by a ring with the inner radius of 50 μm and outer radius of 100 μm from Cartesian to log-polar coordinate. As can be seen in Fig. S15c, the transformed image in Cartesian coordinate is a ring which agrees well with the theoretical calculation result in Fig. S15b.

Figure S15. The log-polar to Cartesian coordinate transformation using metasurfaces to reconstruct the ring-shaped image in Fig. 3 in the main text.

On the other hand, we have transformed airplane-shaped images in Cartesian coordinate to log-polar coordinate using metasurfaces as shown in Fig. 4 in the main text. We can also utilize metasurfaces to transform the images in Figs. 4d-4f back to Figs. 4a-c by performing log-polar to Cartesian coordinate transformation. To this end, we projected the image in Fig. S16a on a metasurface which was encoded by the phase profile in Eq. S30 to perform log-polar to Cartesian coordinate transformation. The image in Fig. S16a has the same shape and location as the one in Fig. 4e in the main text, which was previously transformed by the airplane-shaped image in Fig. 4b from Cartesian to log-polar coordinate. As can be seen in Fig. S16c, although the transformed image in Cartesian coordinate has rough intensity distribution due to the uneven sampling rate during the transformation, it has the same shape as the theoretical calculation result in Fig. S16b.

Figure S16. The log-polar to Cartesian coordinate transformation using metasurfaces to reconstruct airplane images in Fig. 4 in the main text.

”

We have also included the citation of the newly added Supplementary Note 9 in the main text, and copied the corresponding sentences below for the reviewer’s convenience.

The second paragraph on page 12 in the main text now reads:

“

... ...In addition, we can also transform the log-polar image which was previously transformed from an image in the Cartesian coordinate, back to the original image by a log-polar to Cartesian coordinate transforming metasurface (Supplementary Note 9). ...

”

3.6 missing information

Height of Si nanostructure, defined as t , is not shown.

Ref. 4 does not have volume and page.

We thank the reviewer for pointing out these errors. The height of Si nanostructure is 500 nm. This detail has been incorporated into the revised main text on page 6, with the updated content provided below:

“

... Taking advantage of the geometric phase (or Pancharatnam-Berry phase), we utilized amorphous silicon nano-bars with a thickness of 500 nm to convert

incident circular polarized light to its orthogonally polarized counterpart with specific abrupt phase shifts that are linearly dependent on the nano-bars' orientation angles (Figs. 2a and 2b).

”

We have also revised the citation information for Ref. 4 (i.2. Ref. 7 in the revised main text), and it now reads:

“

7. Cordaro A, Edwards B, Nikkhah V, Alu A, Engheta N, Polman A. Solving integral equations in free space with inverse-designed ultrathin optical metagratings. *Nat Nanotechnol* **18**, 365–372 (2023).

”

REVIEWER COMMENTS

Reviewer #1 (Remarks to the Author):

The authors have addressed my questions and comments very well. I am happy with the current version. I think this manuscript can be accepted now.

Reviewer #2 (Remarks to the Author):

I have carefully read the revised manuscript as well as the authors' responses. I appreciate the authors' effort to address my comments and those of the other referees with solid and rigorous data. The manuscript has been significantly improved. I believe it is an important contribution to the field of image processing, machine vision and meta-optics, and recommend its publication in Nature Communications.

Reviewer #3 (Remarks to the Author):

The authors revised the manuscript extensively to reply to all the comments raised by the reviewers.

As far as I examined the manuscript and supplementary information, they addressed all the comments.

The contents have been made clear to me; consequently, one of the main claims by the authors is found to be shape-independent transformation.

Here, one question emerges.

Although they intend to the shape-independent transformations, there is one constraint. That is, the original shapes have to be set in a centrosymmetric manner. It is necessary for the authors to clarify the tolerance regarding the center position. For example, how much offset is allowed for the configurations in Figure 3 and 4.

I guess that 0.1 mm offset is not allowed.

The authors are responsible for addressing the limitation of the transformation to maintain scientific sound in the manuscript.

If the authors could clarify the tolerance in the text, this manuscript would be publishable in Nature Communications. Otherwise, unpublishable.

Point-by-Point Response Letter

We thank the reviewers for their valuable feedback. Each comment is in red, which is addressed directly and as clearly as possible in blue. All the changes we made are highlighted in the main text and are included here in green. Furthermore, we have included additional discussions in the Supplementary Information to support our results.

Responses to Reviewer 1:

The authors have addressed my questions and comments very well. I am happy with the current version. I think this manuscript can be accepted now.

We greatly appreciate the reviewer's recommendation to accept our work.

Responses to Reviewer #2:

I have carefully read the revised manuscript as well as the authors' responses. I appreciate the authors' effort to address my comments and those of the other referees with solid and rigorous data. The manuscript has been significantly improved. I believe it is an important contribution to the field of image processing, machine vision and meta-optics, and recommend its publication in Nature Communications.

We are deeply grateful to the reviewer for the positive feedback, for recognizing the novelty of our work, and for recommending its publication in Nature Communications.

Responses to Reviewer #3:

The authors revised the manuscript extensively to reply to all the comments raised by the reviewers. As far as I examined the manuscript and supplementary information, they addressed all the comments. The contents have been made clear to me; consequently, one of the main claims by the authors is found to be shape-independent transformation.

Here, one question emerges. Although they intend to the shape-independent transformations, there is one constraint. That is, the original shapes have to be set in a centrosymmetric manner. It is necessary for the authors to clarify the tolerance regarding the center position. For example, how much offset is allowed for the configurations in Figure 3 and 4. I guess that 0.1 mm offset is not allowed. The authors are responsible for addressing the limitation of the transformation to maintain scientific sound in the manuscript. If the authors could clarify the tolerance in the text, this manuscript would be publishable in Nature Communications. Otherwise, unpublishable.

We greatly appreciate the reviewer's positive feedback and insightful comments on our work. We also want to thank the reviewer for raising the important point regarding the effect of the input image's position on the geometric transformation. Following the reviewer's suggestions, we believe that our revised manuscript has been considerably strengthened. We have included additional results analyzing the effect of the linear translation on the Cartesian to log-polar coordinate transformation in the newly added Supplementary Note 11.

In essence, the geometric image-transforming metasurface establishes a mapping between the input images and their transformed counterparts through the coordinate transformation relationship. We demonstrated that the scale and rotation operations of the images in Cartesian coordinate are converted to a simple linear translation operation in log-polar coordinate, with the shape of the transformed images in log-polar coordinate remaining consistent. However, it is important to note that the

Cartesian to log-polar coordinate transformation is not translation-invariant. When an input image undergoes a linear translation operation in Cartesian coordinate, its corresponding image in log-polar coordinate also alters. Yet, if the linear translation of the input image is small, the change in the transformed image can be negligible.

To comprehensively study the effect of the input image's position on the shape of the transformed image, we used the Cartesian to log-polar coordinate transformation of airplane shapes demonstrated in Figs. 4-5 of the main text as an example. According to our analysis, the tolerance for the center position of the airplane shapes is roughly around 21 μm , which is about 10% of the input image's width.

In addition, we demonstrated that scale- and rotation- invariances of the log-polar transformation remained valid if input images were scaled and/or rotated about the origin of the Cartesian coordinate, even when the image center positions were not aligned with the Cartesian origin.

We have made corresponding clarification in the main text. The revised last paragraph on page 9 of the main text now reads as follows:

“

To confirm the scale- and rotation-invariance of our metasurfaces for the log-polar transformation, we utilized three airplane-shaped binary images $f_1(x, y)$, $f_2(x, y)$ and $f_3(x, y)$ with different sizes and orientations as the input images for the metasurface (Figs. 4a-4c). We set the center of the shapes as the origin for the Cartesian coordinate (Supplementary Note 11). In our experiment,

”

In addition, the newly added Supplementary Note 11 reads:

“

11. The effect of the input image's linear translation on the Cartesian to log-polar coordinate transformation

For the Cartesian to log-polar coordinate transformation, we have the coordinate relations below:

$$X(x, y) = a \cdot \ln \frac{r}{b}, \quad (\text{S37})$$

$$Y(x, y) = -a \cdot \alpha, \quad (\text{S38})$$

$$r = \sqrt{x^2 + y^2}, \quad (\text{S39})$$

$$\alpha = \text{atan2}(y, x), \quad (\text{S40})$$

Each input image corresponds to a unique transformed image, which is determined solely by the coordinate transformation relationship described in Eqs. S37-S40. When the input image is scaled by $s = \frac{r'}{r}$ and rotated by $\theta = \alpha' - \alpha$, the expressions become:

$$X'(x, y) = a \cdot \ln \frac{r'}{b}, \quad (\text{S41})$$

$$Y'(x, y) = -a \cdot \alpha', \quad (\text{S42})$$

From these, we derive:

$$X' = X + a \cdot \ln s, \quad (\text{S43})$$

$$Y' = Y - a \cdot \theta, \quad (\text{S44})$$

From Eqs. S43-S44, it is clear that the scale and rotation in Cartesian coordinate are transformed into linear translations along X and Y axes in the log-polar coordinate, without changing the shape of the transformed image. We refer to such transformations as scale- and rotation- invariant transformations. Since the scale and rotation operations, determined by $s = \frac{r'}{r}$ and $\theta = \alpha' - \alpha$, are centered around the origin of the Cartesian coordinate, scale- and rotation-invariances are only valid when the input images are scaled and/or rotated about the origin of the Cartesian coordinate.

On the other hand, when the input image undergoes a linear translation in the Cartesian coordinate (*i.e.*, the center position of the input image is shifted to $r' = r - r_0$), we have:

$$X'(x, y) = a \cdot \ln \frac{r-r_0}{b}, \quad (\text{S45})$$

$$Y'(x, y) = -a \cdot \alpha, \quad (\text{S46})$$

From these, we can get:

$$X' = X + a \cdot \ln\left(1 - \frac{r_0}{r}\right), \quad (\text{S47})$$

$$Y' = Y, \quad (\text{S48})$$

From Eqs. S47-S48, it is evident that the coordinate of the new transformed image along the Y-axis remains the same as that of the original input image. However, the coordinate of the new transformed image along X-axis is translated by $a \cdot \ln\left(1 - \frac{r_0}{r}\right)$, which is not a constant. Thus, when the input image undergoes linear translation in Cartesian coordinates, the Cartesian to log-polar coordinate transformation is no longer invariant.

To comprehensively study the effect of the position of the input image's position on the shape of the transformed image, we use the airplane shape illustrated in Figs. 4-5 of the main text as an example. In this case, the geometric parameters of the airplane are represented by (s, r, t) , where s , r and t denote the scale, rotation angle and linear translation of the shape. For the airplane shapes in depicted Fig. 4 of the main text, the linear translation was set to zero ($t = 0$). We re-plotted their Cartesian to log-polar coordinate transformations in Fig. S22. Evidently, the transformations remained scale- and rotation-invariant when the linear translation was zero.

Figure S22. The scale- and rotation- invariances of the log-polar coordinate transformation using metasurfaces. The log-polar coordinate transformation for input airplane shapes $f(x, y)$ in the Cartesian coordinate, each with a different scale factor (s) and a rotation angle (r) but zero linear translation ($t=0$). $f_s(x, y)$: $s=1.0$, $r=0$, $t=0$ (a); $f_0(x, y)$: $s=1.5$, $r=0$, $t=0$ (b); $f_r(x, y)$: $s=1.5$, $r=\pi/6$, $t=0$ (c). Top: the white and black regions of the input images indicate the transmittance of 100% and 0%, respectively. Bottom: the mathematical and simulated transformed images in the log-polar coordinate for the input airplane shapes.

We then studied the effect of linear translation on the Cartesian to log-polar coordinate transformation. In this case, we chose $f_0(x, y)$ with a scaling factor of 1.5 and a rotation angle of 0 as an example. Figure S23a shows that the autocorrelation for $g_0(X, Y)$ exhibited a bright spot at the origin, indicating a perfect match. As the linear translation along the $-X$ axis increased, the distortion of the transformed image became larger, evidenced by the larger spot in the cross-correlation maps. Thus, as predicted in Eqs. S47-S48, we can conclude that the Cartesian to log-polar coordinate transformation is not translation-invariant. Nevertheless, regarding the log-polar transformation of $f_1(x, y)$ with a small linear translation, for example $21 \mu\text{m}$, the cross-correlation with $g_0(X, Y)$ still displayed a main bright spot in Fig. S23b. When the linear translation exceeded $21 \mu\text{m}$, the cross-correlation maps showed expanded bright regions, indicating poor similarities (Figs. S23c-S23e). Therefore, the tolerance for the center position shift of our airplane shapes is approximately $21 \mu\text{m}$, which constitutes about 10% of the shape's width.

Figure S23. The effect of the input image's linear translation on the Cartesian to log-polar coordinate transformation using metasurfaces. The log-polar coordinate transformation for input airplane shapes $f(x, y)$ in the Cartesian coordinate, each with a different linear translation (t). $f_0(x, y)$: $s=1.5$, $r=0$, $t=0$ (a); $f_1(x, y)$: $s=1.5$, $r=0$, $t=-21 \mu\text{m}$ (b); $f_2(x, y)$: $s=1.5$, $r=0$, $t=-42 \mu\text{m}$ (c); $f_3(x, y)$: $s=1.5$, $r=0$, $t=-63 \mu\text{m}$ (d); $f_4(x, y)$: $s=1.5$, $r=0$, $t=-84 \mu\text{m}$ (e). Top: The white and black regions of the input images indicate the transmittance of 100% and 0%, respectively. Middle: the mathematical and simulated transformed images in the log-polar coordinate corresponding to the input airplane shapes. Bottom: the correlation maps with respect to $g_0(X, Y)$.

After characterizing the effect of linear translation on the Cartesian to log-polar coordinate transformation, we further explored the scale-variances for the log-polar transformations of images with mismatched center positions. We compared $f_5(x, y)$ with $f_0(x, y)$, $f_1(x, y)$, $f_2(x, y)$, $f_3(x, y)$, and $f_4(x, y)$. For these airplane images, the scale factor for $f_5(x, y)$ was 1.0, while it was 1.5 for $f_0(x, y)$, $f_1(x, y)$, $f_2(x, y)$, $f_3(x, y)$, and $f_4(x, y)$. The scaling of airplane images was relative to the center of the shape. Additionally, the linear translation for $f_5(x, y)$ was zero, while the linear translations for $f_0(x, y)$, $f_1(x, y)$, $f_2(x, y)$, $f_3(x, y)$ and $f_4(x, y)$ were increased from 0 to $-84 \mu\text{m}$ in increments of $-21 \mu\text{m}$. We then performed the log-polar transformation for these airplane images using the metasurface. We then calculated the correlations

between $g_s(X, Y)$ and $g_s(X, Y)$, $g_0(X, Y)$, $g_1(X, Y)$, $g_2(X, Y)$, $g_3(X, Y)$, $g_4(X, Y)$, respectively. As shown in Fig. S24, the autocorrelation of $g_s(X, Y)$ displayed a bright spot at the origin, indicating a perfect match. The cross-correlation between $g_s(X, Y)$ and $g_0(X, Y)$ showed a shifted bright spot due to the scale-invariance of log-polar transformation. However, from $g_1(X, Y)$ to $g_4(X, Y)$, the cross-correlation respect to $g_s(X, Y)$ exhibited broadened bright regions, indicating poor similarities. Therefore, for airplane images with mismatched center positions, the Cartesian to log-polar coordinate transformation was not scale-invariant when images were scaled about their individual center positions.

Figure S24. The scale-variances for the log-polar transformations of images with mismatched center positions. The images were scaled about the center positions of the airplane shapes. The correlations between $g_s(X, Y)$ in Fig. S22 and $g_s(X, Y)$ (a), $g_0(X, Y)$ (b), $g_1(X, Y)$ (c), $g_2(X, Y)$ (d), $g_3(X, Y)$ (e), $g_4(X, Y)$ (f) in Fig. S23.

Similarly, we also examined the rotation-variances for the log-polar transformations of images with mismatched center positions. We compared $f_r(x, y)$ with $f_0(x, y)$, $f_1(x, y)$, $f_2(x, y)$, $f_3(x, y)$, and $f_4(x, y)$. For these airplane images, the rotation angle for $f_r(x, y)$ was $\pi/6$, while it was 0 for $f_0(x, y)$, $f_1(x, y)$, $f_2(x, y)$, $f_3(x, y)$, and $f_4(x, y)$. The rotation of the airplane images was relative to the shape's center. Additionally, the linear translation for $f_r(x, y)$ was zero, while the linear translations for $f_0(x, y)$, $f_1(x, y)$, $f_2(x, y)$, $f_3(x, y)$, and $f_4(x, y)$ were increased from 0 to $-84 \mu\text{m}$ in increments of $-21 \mu\text{m}$. We then performed the log-polar transformation for these airplane images using the metasurface. We then calculated the correlations

between $g_r(X, Y)$ and $g_r(X, Y)$, $g_0(X, Y)$, $g_1(X, Y)$, $g_2(X, Y)$, $g_3(X, Y)$, $g_4(X, Y)$, respectively. As shown in Fig. S25, the autocorrelation of $g_r(X, Y)$ displayed a bright spot in the origin, indicating a perfect match. The cross-correlation between $g_r(X, Y)$ and $g_0(X, Y)$ showed a shifted bright spot due to the rotation-invariance of log-polar transformation. However, from $g_1(X, Y)$ to $g_4(X, Y)$, the cross-correlation respect to $g_r(X, Y)$ exhibited broadened bright regions, indicating poor similarities. Therefore, for airplane images with mismatched center positions, the Cartesian to log-polar coordinate transformation was not rotation-invariant when images were rotated about their individual center positions.

Figure S25. The rotation-variances for the log-polar transformations of images with mismatched center positions. The images were rotated about the center positions of the airplane shapes. The correlations between $g_r(X, Y)$ in Fig. S22 and $g_r(X, Y)$ (a), $g_0(X, Y)$ (b), $g_1(X, Y)$ (c), $g_2(X, Y)$ (d), $g_3(X, Y)$ (e), $g_4(X, Y)$ (f) in Fig. S23.

Nevertheless, if considering the log-polar transformation of $f_1(x, y)$ with a small linear translation of 21 μm , the cross-correlations with $g_c(X, Y)$ and $g_r(X, Y)$ both exhibited a single main bright spot in Figs. S24c and S25c, respectively. When the linear translation exceeded 21 μm , the cross-correlation maps revealed broader bright regions, suggesting poor similarities (Figs. S24d-S24f and S25d-S25f). Therefore, the tolerance for the center position of the airplane shapes is approximately 21 μm , which constitutes about 10% of the shape's width.

Figure S26. The scale- and rotation- invariances for the log-polar transformations when the center positions of images were not at the origin of the Cartesian coordinate. Top: airplane shapes in the Cartesian coordinate with different scales and rotation angles. $f_{2s}(x, y)$: $s=1.0$, $r=0$ (a); $f_2(x, y)$: $s=1.5$, $r=0$ (b); $f_{2r}(x, y)$: $s=1.5$, $r=\pi/6$ (c). The white and black regions for the input airplane shapes indicate the transmittance of 100% and 0%, respectively. The scale and rotation of the images were about the origin of the Cartesian coordinate. Middle: the mathematical and simulation transformed images in the log-polar coordinate for the input airplane shapes. Bottom: the correlations respect to $g_2(X, Y)$.

Finally, we explored the log-polar transformations for the images that underwent scale and rotation scale and rotation operations relative to the origin of the Cartesian coordinate, even when the image center positions were not aligned with the Cartesian origin. We considered $f_2(x, y)$ whose center position was $42 \mu\text{m}$ away from the origin.

After scale and rotation operations with respect to the origin, we obtained $f_{2s}(x, y)$ and $f_{2r}(x, y)$ (Fig. S26). We then performed log-polar coordinate transformation for $f_2(x, y)$, $f_{2s}(x, y)$, and $f_{2r}(x, y)$, resulting in $g_2(X, Y)$, $g_{2s}(X, Y)$, and $g_{2r}(X, Y)$. To assess the similarity among $g_2(X, Y)$, $g_{2s}(X, Y)$, and $g_{2r}(X, Y)$, we conducted a correlation analysis. As shown in Fig. S26, all the three correlation maps $R_{g_2g_2}$, $R_{g_2g_{2s}}$ and $R_{g_2g_{2r}}$ displayed single small bright spot, indicating perfect matches. Therefore, if the scaling and rotation of images were about the origin of the Cartesian coordinate, the log-polar transformations of the images remain scale- and rotation-invariant, even when the image center positions were not aligned with the Cartesian origin.

”

REVIEWERS' COMMENTS

Reviewer #3 (Remarks to the Author):

The authors revised the point on which I raised a question.

The tolerance is not specified in the text though they mentioned it in the supplementary information that becomes very long now. I hope that readers could find the point, otherwise readers may be misled.

Through this revision, it turns out that the transformation proposed by the authors is invariant, only if the center of the original image is set to the center quite precisely. This would be a problem in a practical application because such a precise alignment is usually cost-demanding.

As a research article, I think that this manuscript has become publishable.

Point-by-Point Response Letter

We thank the reviewers for their valuable feedback. Each comment is in red, which is addressed directly and as clearly as possible in blue. All the changes we made are highlighted in the main text and are included here in green.

Responses to Reviewer 3:

The authors revised the point on which I raised a question. The tolerance is not specified in the text though they mentioned it in the supplementary information that becomes very long now. I hope that readers could find the point, otherwise readers may be misled.

Through this revision, it turns out that the transformation proposed by the authors is invariant, only if the center of the original image is set to the center quite precisely. This would be a problem in a practical application because such a precise alignment is usually cost-demanding.

As a research article, I think that this manuscript has become publishable.

We sincerely appreciate the reviewer's recommendation to accept our work. Following the reviewer's suggestion, we have included the tolerance into the main text. The revised content can be found in the first paragraph on page 10 of the main text.

“

... .. As the log-polar transformation is not translation-invariant, and the tolerance for the center position of the airplane shapes is about 10% of the input image's width, we set the center of the shapes as the origin for the Cartesian coordinate to avoid effects from translation (Supplementary Note 11).

”